# Overcoming Lower-Level Constraints in Bilevel Optimization: A Novel Approach with Regularized Gap Functions

**Wei Yao**[1,2] **Haian Yin**[2] **Shangzhi Zeng**[1,2] **Jin Zhang**[2,1,3*]

[1]National Center for Applied Mathematics Shenzhen
[2]Department of Mathematics, Southern University of Science and Technology
[3]CETC Key Laboratory of Smart City Modeling Simulation and Intelligent Technology, The Smart City Research Institute of CETC
[*]Corresponding author(s). E-mail(s): `zhangj9@sustech.edu.cn`;
Contributing authors: `yaow@sustech.edu.cn`; `yinha@sustech.edu.cn`; `zengsz@sustech.edu.cn`;

## Abstract

Constrained bilevel optimization tackles nested structures present in constrained learning tasks like constrained meta-learning, adversarial learning, and distributed bilevel optimization. However, existing bilevel optimization methods mostly are typically restricted to specific constraint settings, such as linear lower-level constraints. In this work, we overcome this limitation and develop a new single-loop, Hessian-free constrained bilevel algorithm capable of handling more general lower-level constraints. We achieve this by employing a doubly regularized gap function tailored to the constrained lower-level problem, transforming constrained bilevel optimization into an equivalent single-level optimization problem with a single smooth constraint. We rigorously establish the non-asymptotic convergence analysis of the proposed algorithm under the convexity of lower-level problem, avoiding the need for strong convexity assumptions on the lower-level objective or coupling convexity assumptions on lower-level constraints found in existing literature. Additionally, the generality of our method allows for its extension to bilevel optimization with minimax lower-level problem. We evaluate the effectiveness and efficiency of our algorithm on various synthetic problems, typical hyperparameter learning tasks, and generative adversarial network.

## 1 Introduction

Bilevel optimization (BiO), which subsumes minimax optimization as a special case, is a hierarchical optimization comprising two levels, with one problem nested within another. There is growing interest in BiO that is driven by an abundance of applications. Examples in machine learning (ML) include hyperparameter optimization (Pedregosa, 2016; Franceschi et al., 2018; Mackay et al., 2019), meta-learning (Franceschi et al., 2018; Zügner & Günnemann, 2018; Rajeswaran et al., 2019; Ji et al., 2020), and reinforcement learning (Kunapuli et al., 2008; Hong et al., 2023). Typically, the lower-leve (LL) problems of BiO in ML literature handle learning tasks as unconstrained optimization problems. However, constraints are crucial for learning tasks and are becoming increasingly important in designing robust, fair, and safe ML systems (Silva & Najafirad, 2020; Mehrabi et al., 2021; Yang et al., 2023). The resulting constrained bilevel problem applies to a wider range of applications than the unconstrained one, such as adversarial learning (Madry et al., 2018; Wong et al., 2019; Zhang et al., 2022), federated learning (Fallah et al., 2020; Tarzanagh et al., 2022; Yang et al., 2024), continual learning (Lopez-Paz & Ranzato, 2017), and meta-learning for few-shot learning (Xu & Zhu, 2023).

Existing BiO methods mainly focus on the LL unconstrained case (Franceschi et al., 2018; Ghadimi & Wang, 2018; Grazzi et al., 2020; Ji et al., 2021; Chen et al., 2021; Ji et al., 2022; Hong et al., 2023; Arbel & Mairal, 2022; Dagréou et al., 2022; Kwon et al., 2023; Huang, 2023). Recently, a few approaches have emerged to tackle BiO problems with constraints. However, most of these

approaches are limited to linear constraints or constraints that rely solely on the LL variable. In this study, we address more general constrained BiO problems, where the LL problems involve constraints coupling both upper-level (UL) and LL variables. Explicitly, we consider the constrained bilevel optimization problem of the following form:

$$\min_{x \in X, y \in Y} F(x, y) \quad \text{s.t.} \quad y \in S(x), \tag{1}$$

with $S(x)$ being the set of optimal solutions of the constrainted lower-level problem,

$$\min_{y \in Y} f(x, y) \quad \text{s.t.} \quad g(x, y) \leq 0, \tag{2}$$

where $x \in \mathbb{R}^n$, $y \in \mathbb{R}^m$, $X$ and $Y$ are closed convex sets in $\mathbb{R}^n$ and $\mathbb{R}^m$, respectively. The UL objective $F : X \times Y \rightarrow \mathbb{R}$, the LL objective $f : X \times Y \rightarrow \mathbb{R}$, and the LL constraint mapping $g : X \times Y \rightarrow \mathbb{R}^p$ are continuously differentiable functions.

The constrained BiO problem (1) is challenging due to the implicit constraint $y \in S(x)$, which requires $y$ to be a solution of an optimization problem. The LL constraints introduce extra complexity, especially when dealing with coupled LL constraints. This complexity hinders the straightforward extension of existing methods to constrained BiO problems (Kwon et al., 2024).

In the pursuit of solving constrained BiO problem, several algorithms have been developed to address specific cases of (1). For instance, the works (Tsaknakis et al., 2022; Khanduri et al., 2023; Kornowski et al., 2024) handle constrained BiO problems with $g(x, y) = Ay - b$ and $f(x, y)$ being strong convexity with respect to (*w.r.t.*) $y$. As for coupling constrained LL problem, Xiao et al. (2023) introduces an alternating projected SGD approach for a family of BiO problems with coupling LL linear equality constraints $Ay + h(x) - c = 0$, where $A$ represents a matrix, $c$ denotes a vector, and $h(x)$ is a vector-valued function. Beyond the confines of a specific setting, Xu & Zhu (2023) develops a gradient-based approach for general BiO problems, where the LL problem is convex with equality and inequality constraints. Since the methods presented in these works predominantly utilize implicit gradient-based techniques, it is unexpected that they all presuppose the strong convexity of the LL objective *w.r.t.* $y$, alongside other regularity conditions, to ensure the uniqueness and smoothness of the LL solution mapping. More importantly, the algorithms in these studies would tolerate the heavier memory and computational cost of using second-order calculations in large-scale applications.

Therefore, it is imperative to develop first-order methods that do not necessitate explicit estimation of the implicit gradients, thereby facilitating the resolution of a broader class of constrained BiO problems. In pursuit of this goal, the value function approach has garnered considerable attention due to its efficacy in designing first-order, single-loop numerical algorithms (Ye & Zhu, 1995; Kwon et al., 2023; Shen & Chen, 2023). However, challenges arise when the LL problem involves constraints. Specifically, the value function, $v(x) := \min_{y \in Y}\{f(x, y) \,|\, g(x, y) \leq 0\}$, is usually an implicit nonsmooth function, even when the underlying problem functions possess favorable properties. This difficulty is further exacerbated in the presence of coupled LL constraints, where $v(x)$ is typically nonsmooth, making it challenging to compute or approximate its generalized gradient. Recently, Jiang et al. (2024) proposed a primal-dual-assisted penalty approach to address these challenges associated with coupled LL constraints, based on the value function reformulation.

A notable distinctive approach for addressing general constrained BiO problems has recently been presented in Yao et al. (2024), which introduces a novel proximal Lagrangian value function to tackle constrained LL problems. By utilizing this function, they convert the constrained BiO problem into an equivalent optimization problem with smooth constraints. Notably, their reformulation preserves the coupling LL constraints, akin to the value function approach. Consequently, the proposed algorithm (LV-HBA) in Yao et al. (2024) involves Euclidean projection onto the coupled LL constraint set. However, such an operation typically requires the assumption of coupling convexity and can be costly when the set is complex. Therefore, a natural question arises: *Can we develop a first-order algorithm to overcome possibly coupled lower-level constraints in bilevel optimization?*

Our response to this question is affirmative. Next, we highlight the main contributions of this study. Additional related work is provided in Appendix A.1.

- **Reducing constrained bilevel optimization into an equivalent optimization problem with only a single smooth inequality constraint.** We propose a novel single-level smoothed reformulation for constrained BiO with possibly coupled LL constraints. A key

technique is the doubly regularized gap function defined in (3), which serves as an optimality metric for the LL problems and allows for straightforward Hessian-free gradient evaluation, as shown in (4). Furthermore, this type of gap function can be readily extended to tackle more complex bilevel optimization scenarios involving minimax lower-level problems.

- **Developing a single-loop first-order algorithm without projection onto the coupled lower-level constraint set.** Building upon the newly introduced single-leve reformulation, we propose **Bi**level **C**onstrained **GA**p **F**unction-based **F**irst-order **A**lgorithm (**BiC-GAFFA**), a first-order algorithm that can be implemented entirely within a single-loop framework. Furthermore, we rigorously establish the non-asymptotic convergence analysis of BiC-GAFFA under the convexity of LL problem, avoiding the necessity for either the strong convexity assumption on the LL objective or the full convexity assumption on the LL constraints.
- We validate the effectiveness and efficiency of BiC-GAFFA on various synthetic problems, typical hyper-parameter learning tasks, and generative adversarial network. These experiments collectively substantiate the superior performance of BiC-GAFFA.

## 2 REGULARIZED GAP FUNCTION AND EQUIVALENT REFORMULATION

Transforming a BiO problem to a single-level optimization problem is a useful strategy from both theoretical and computational perspectives. In this section, we introduce a novel smoothed reformulation tailored for constrained BiO problems with potentially coupled LL constraints. For this purpose, we define the following doubly regularized gap function for the LL problem (2):

$$\mathcal{G}_\gamma(x,y,z) := \max_{\theta \in Y, \lambda \in \mathbb{R}_+^p} \left\{ \mathcal{L}(x,y,\lambda) - \frac{1}{2\gamma_2}\|\lambda - z\|^2 - \mathcal{L}(x,\theta,z) - \frac{1}{2\gamma_1}\|\theta - y\|^2 \right\}, \quad (3)$$

where $z \in \mathbb{R}^p$, the Lagrangian function $\mathcal{L}(x,y,z) := f(x,y) + z^{\mathrm{T}}g(x,y)$, and $\gamma := (\gamma_1, \gamma_2) > 0$ is the regularization (or proximal) parameters. This regularized gap function concept has been previously applied in various contexts, such as variational inequalities (Fukushima, 1992), standard Nash games (Gürkan & Pang, 2009), and the generalized Nash equilibrium problem (Von Heusinger & Kanzow, 2009). More recently, it has been successfully utilized in addressing optimal control problems with equilibrium constraints (Lin & Ohtsuka, 2024). However, its application in studying BiO problems with constrained lower-level problem remains unexplored.

When the LL problem is convex, and the associated multipliers exist for any $y \in S(x)$, that is, $\mathcal{M}(x,y) := \left\{ \lambda \in \mathbb{R}_+^p \mid 0 \in \nabla_y f(x,y) + \lambda^{\mathrm{T}}\nabla_y g(x,y) + \mathcal{N}_Y(y), \ \lambda^{\mathrm{T}}g(x,y) = 0 \right\} \neq \varnothing$, where $\mathcal{N}_Y(y)$ denotes the normal cone of $Y$ at point $y$, i.e., $\mathcal{N}_Y(y) := \{v \in \mathbb{R}^m \mid \langle v, y' - y \rangle \leq 0, \ \forall y' \in Y\}$, the single inequality constraint $\mathcal{G}_\gamma(x,y,z) \leq 0$ can equivalently characterizes the solution set of LL problem.

**Lemma 2.1.** *Assume that both $f(x,\cdot)$ and $g(x,\cdot)$ are convex for any $x \in X$. Let $\gamma_1, \gamma_2 > 0$, we have $\mathcal{G}_\gamma(x,y,z) \geq 0$ for any $(x,y,z) \in X \times Y \times \mathbb{R}_+^p$. Furthermore, for any $(x,y,z) \in X \times Y \times \mathbb{R}_+^p$, $\mathcal{G}_\gamma(x,y,z) \leq 0$ if and only if $y \in S(x)$ and $z \in \mathcal{M}(x,y)$.*

Another advantageous property of $\mathcal{G}_\gamma$ is its continuously differentiability when both functions $f$ and $g$ exhibit continuous differentiability.

**Lemma 2.2.** *Assume that both $f(x,y)$ and $g(x,y)$ are convex in $y$ on $Y$ for any $x \in X$ and be continuously differentiable on an open set containing $X \times Y$. Then $\mathcal{G}_\gamma(x,y,z)$ is continuously differentiable on $X \times Y \times \mathbb{R}_+^p$, and for any $(x,y,z) \in X \times Y \times \mathbb{R}_+^p$,*

$$\nabla \mathcal{G}_\gamma(x,y,z) = \begin{pmatrix} \nabla_x f(x,y) + (\lambda^*)^{\mathrm{T}}\nabla_x g(x,y) \\ \nabla_y f(x,y) + (\lambda^*)^{\mathrm{T}}\nabla_y g(x,y) \\ -(z - \lambda^*)/\gamma_2 \end{pmatrix} - \begin{pmatrix} \nabla_x f(x,\theta^*) + z^{\mathrm{T}}\nabla_x g(x,\theta^*) \\ (y - \theta^*)/\gamma_1 \\ g(x,\theta^*) \end{pmatrix}, \quad (4)$$

*where $\theta^*$ and $\lambda^*$ denote $\theta^*(x,y,z)$ and $\lambda^*(x,y,z)$, respectively, defined as*

$$\theta^*(x,y,z) := \underset{\theta \in Y}{\operatorname{argmin}} \left\{ f(x,\theta) + z^{\mathrm{T}}g(x,\theta) + \frac{1}{2\gamma_1}\|\theta - y\|^2 \right\},$$

$$\lambda^*(x,y,z) := \underset{\lambda \in \mathbb{R}_+^p}{\operatorname{argmax}} \left\{ f(x,y) + \lambda^{\mathrm{T}}g(x,y) - \frac{1}{2\gamma_2}\|\lambda - z\|^2 \right\} = \operatorname{Proj}_{\mathbb{R}_+^p}\left(z + \gamma_2 g(x,y)\right). \quad (5)$$

Now we derive a smooth single-level reformulation for the constrained BiO problem (1):

$$\min_{(x,y,z)\in X\times Y\times \mathbb{R}^p_+} F(x,y) \quad \text{s.t.} \quad \mathcal{G}_\gamma(x,y,z) \leq 0. \tag{6}$$

This reformulation problem (6) is equivalent to the original BiO problem (1).

**Theorem 2.3.** *Assume that both $f(x,\cdot)$ and $g(x,\cdot)$ are convex for any $x \in X$. Let $\gamma_1, \gamma_2 > 0$, the reformulation (6) is equivalent to the bilevel optimization problem (1), provided that for any feasible point $(x,y)$ of (1), a corresponding multiplier of the lower-level problem (2) exists at $(x,y)$, i.e., $\mathcal{M}(x,y) \neq \varnothing$. Specifically, $(x^*,y^*)$ is an optimal solution to the bilevel optimization problem (1) and $z^* \in \mathcal{M}(x^*,y^*)$ if and only if $(x^*,y^*,z^*)$ is an optimal solution to the reformulation (6).*

*Remark* 2.4. The equivalent reformulation (6) possesses two noteworthy characteristics. First, the formulation includes only one inequality constraint. Second, the LL constraints $g(x,y) \leq 0$ are not explicitly stated in the reformulation (6), in contrast to the value function-based reformulation as well as the formulation presented in (Yao et al., 2024). Note also that the proofs of Lemmas and Theorem presented in this section are available in Appendix A.3.

*Remark* 2.5. The assumption of the existence of the multiplier of the lower-level problem (2) in Theorem 2.3 can be guaranteed when the lower-level constraint $g(x,y) \leq 0$ satisfies a constraint qualification condition such as the Guignard's CQ, linear independence constraint qualification (LICQ), or Mangasarian-Fromovitz constraint qualification (MFCQ).

## 3 GAP FUNCTION-BASED FIRST-ORDER ALGORITHM

Building upon the newly introduced reformulation, we proceed to the algorithm design phase.

First, to enhance the stability of the proposed numerical algorithms, we introduce an upper bound constraint on the variable $z$. Consequently, we consider the following truncated variant of the reformulation (6):

$$\min_{(x,y,z)\in X\times Y\times Z} F(x,y) \quad \text{s.t.} \quad \mathcal{G}_\gamma(x,y,z) \leq 0, \tag{7}$$

where $Z := [0,r]^p$ with $r \geq 0$ is a compact subset of $\mathbb{R}^p_+$. It is demonstrated that if $r$ is appropriately chosen such that a solution $(x^*,y^*,z^*)$ to (6) satisfies $\|z^*\|_\infty \leq r$, then any optimal solution to (7) is also an optimal solution to the original reformulation (6).

**Proposition 3.1.** *Suppose $\gamma_1, \gamma_2 > 0$ and that an optimal solution $(x^*,y^*,z^*)$ to (6) exists with $z^* \in Z$, then any optimal solution of (7) is also optimal for the reformulation (6).*

Second, to develop a gradient-based algorithm for solving (7), we explore its penalty formulation:

$$\min_{(x,y,z)\in X\times Y\times Z} F(x,y) + c\,\mathcal{G}_\gamma(x,y,z), \tag{8}$$

where $c > 0$ is a penalty parameter. This work focuses on first-order penalty methods for the constrained BiO problem (1). To this end, we propose algorithms to find approximate stationary solutions of (8), in alignment with several previous works (Liu et al., 2021; 2023a; Shen & Chen, 2023; Kwon et al., 2024).

Third, we study the relationship between the penalty formulation (8) and a relaxed problem of (7):

$$\min_{(x,y,z)\in X\times Y\times Z} F(x,y) \quad \text{s.t.} \quad \mathcal{G}_\gamma(x,y,z) \leq \varepsilon, \tag{9}$$

where $\varepsilon > 0$ is the relaxation parameter. The relation between the penalized and relaxed problems of BiO problems has been previously studied in (Shen & Chen, 2023) across various single-level reformulations. Herein, we discuss the relationship between (8) and (9).

**Proposition 3.2.** *Assume that $F(x,y)$ is bounded below by $\underline{F}$ on $X \times Y$. For any $\varepsilon > 0$, there exists $\bar{c} > 0$ such that any global solution $(x_c, y_c, z_c)$ to the penalty formulation (8) with penalty parameter $c \geq \bar{c}$ is also a global solution to the relaxed problem (9) with some relaxation parameter $\varepsilon_c$ satisfying $\varepsilon_c \leq \varepsilon$. Moreover, if $(x_c, y_c, z_c)$ is a local solution to the penalty formulation (8), then it is also a local solution to the relaxed problem (9) with relaxation parameter $\varepsilon_c := \mathcal{G}_\gamma(x_c, y_c, z_c)$.*

If we consider the penalty formulation (8) with an increasing sequence of penalty parameter $c_k$ such that $c_k \to \infty$, any accumulation point of the sequence of solutions associated penalty formulation problem (15) with varying values of $c_k$ is a solution to the problem (7).

**Theorem 3.3.** *Assume that $X$ and $Y$ are closed, and functions $F$, $f$ and $g$ are continuous on $X \times Y$. Suppose that $c_k \to \infty$ and let*

$$(x_k, y_k, z_k) \in \underset{(x,y) \in X \times Y \times Z}{\operatorname{argmin}} F(x, y) + c_k \, \mathcal{G}_\gamma(x, y, z).$$

*Then, any accumulation point $(\bar{x}, \bar{y}, \bar{z})$ of the sequence $\{(x_k, y_k, z_k)\}$ is a solution to problem (7).*

The proofs of the lemmas and theorem presented in this section are provided in Appendix A.4.

*Remark* 3.4. The truncation technique applied to the multiplier of lower-level problem in reformulation problems (7) and (8) is commonly used in the literature on numerical augmented Lagrangian methods for nonlinear program (see, e.g., Andreani et al. (2008)). Applying this truncation technique can enhance the stability of the proposed numerical algorithms. As demonstrated in Proposition 3.1, theoretically, if the bound $r$ is selected such that a solution $(x^*, y^*, z^*)$ to (6) exists with $\|z^*\|_\infty \leq r$, then the optimal solution of (7) will also be optimal to (6). Propositions 3.1-3.2 and Theorem 3.3, along with the numerical experiments in Section 6, illustrate that the penalized problem (8) serves as an effective surrogate for the original bilevel problem.

## 3.1 THE PROPOSED ALGORITHM

Now we introduce a first-order gradient-based, single-loop algorithm to solve the truncated and penalized approximation problem (8) with a potentially varying penalty parameter $c_k$. Note that solving (8) still requires care since it involves an optimal value function $\mathcal{G}_\gamma(x, y, z)$ for another optimization problem.

Lemma 2.2 establishes that $\mathcal{G}_\gamma$ is continuously differentiable, implying that the objective function of (8) is also continuously differentiable. Consequently, we can apply a gradient descent-type method to solve (8). However, as also noted in Lemma 2.2, computing the gradient $\nabla \mathcal{G}_\gamma(x^k, y^k, z^k)$ at the current iterate $(x^k, y^k, z^k)$ requires solving the minimization problem in (5) to obtain the exact solution $\theta^*(x^k, y^k, z^k)$, a process which can be computationally intensive.

To mitigate this computational challenge, we introduce an auxiliary sequence $\{\theta^k\}$ as an approximation to $\theta^*$. At each iteration, we employ a single projected gradient descent step to update $\theta^{k+1}$ to approximate $\theta^*(x^k, y^k, z^k)$, as follows:

$$\theta^{k+1} = \operatorname{Proj}_Y \left( \theta^k - \eta_k d_\theta^k \right), \tag{10}$$

where $\eta_k > 0$ is the step size, and

$$d_\theta^k := \nabla_y f(x^k, \theta^k) + (z^k)^{\mathrm{T}} \nabla_y g(x^k, \theta^k) + \frac{1}{\gamma_1}(\theta^k - y^k). \tag{11}$$

Furthermore, we introduce iterate $\lambda^{k+1}$ to represent $\lambda^*(x^k, y^k, z^k)$ as follows:

$$\lambda^{k+1} = \lambda^*(x^k, y^k, z^k) = \operatorname{Proj}_{\mathbb{R}_+^p} \left( z^k + \gamma_2 g(x^k, y^k) \right). \tag{12}$$

By substituting $(\theta^{k+1}, \lambda^{k+1})$ for $(\theta^*, \lambda^*)$ in (4), we can approximate the gradients of the objective function in (8) to define the update directions:

$$
\begin{aligned}
d_x^k := & \frac{1}{c_k} \nabla_x F(x^k, y^k) \\
& + \nabla_x f(x^k, y^k) + (\lambda^{k+1})^{\mathrm{T}} \nabla_x g(x^k, y^k) - \nabla_x f(x^k, \theta^{k+1}) - (z^k)^{\mathrm{T}} \nabla_x g(x^k, \theta^{k+1}), \\
d_y^k := & \frac{1}{c_k} \nabla_y F(x^k, y^k) + \nabla_y f(x^k, y^k) + (\lambda^{k+1})^{\mathrm{T}} \nabla_y g(x^k, y^k) - (y^k - \theta^{k+1})/\gamma_1, \\
d_z^k := & -(z^k - \lambda^{k+1})/\gamma_2 - g(x^k, \theta^{k+1}).
\end{aligned} \tag{13}
$$

Finally, we implement an update for the variables $(x, y, z)$ using a step size $\alpha_k > 0$:

$$(x^{k+1}, y^{k+1}, z^{k+1}) = \operatorname{Proj}_{X \times Y \times Z} \left( (x^k, y^k, z^k) - \alpha_k(d_x^k, d_y^k, d_z^k) \right). \tag{14}$$

The complete algorithm is presented in Algorithm 1. Notably, our proposed algorithm uses only the gradient information of the problem's functions and can be easily implemented when the projections onto the sets $X$ and $Y$ are computationally simple. Furthermore, the updates of $x^{k+1}$, $y^{k+1}$ and $z^{k+1}$ can be executed in parallel, enhancing computational efficiency.

---

**Algorithm 1** **Bi**level **C**onstrained **GA**p Function-based **F**irst-order **A**lgorithm (**BiC-GAFFA**)

---

**Input:** $(x^0, y^0, z^0) \in X \times Y \times Z, \theta^0 \in Y$, stepsizes $\alpha_k, \eta_k > 0$, proximal parameter $\gamma = (\gamma_1, \gamma_2)$, penalty parameter $c_k$;
**for** $k = 0$ **to** $K - 1$ **do**
  - calculate $d_\theta^k$ as in (11);
  - update

$$\theta^{k+1} = \text{Proj}_Y \left( \theta^k - \eta_k d_\theta^k \right), \quad \lambda^{k+1} = \text{Proj}_{\mathbb{R}_+^p} \left( z^k + \gamma_2 g(x^k, y^k) \right);$$

  - calculate $d_x^k, d_y^k, d_z^k$ as in (13);
  - update

$$(x^{k+1}, y^{k+1}, z^{k+1}) = \text{Proj}_{X \times Y \times Z} \left( (x^k, y^k, z^k) - \alpha_k (d_x^k, d_y^k, d_z^k) \right).$$

**end for**

---

## 4 Non-asymptotic convergence analysis

In this section, we rigorously establish the non-asymptotic analysis for BiC-GAFFA towards the truncated and penalized approximation problem

$$\min_{(x,y,z) \in X \times Y \times Z} \psi_c(x, y, z) := F(x, y) + c \, \mathcal{G}_\gamma(x, y, z). \tag{15}$$

The proofs of Lemmas and Theorem presented in this section are available in Appendix A.5.

### 4.1 General assumptions

The following assumptions formalize the smoothness property of the UL objective $F$, and smoothness and convexity properties of the LL objective $f$ and the LL constraints $g$.

**Assumption 4.1** (**UL objective**). The UL objective $F$ is $L_F$-smooth on $X \times Y$. Additionally, $F$ is bounded below on $X \times Y$, *i.e.*, $\underline{F} := \inf_{(x,y) \in X \times Y} F(x, y) > -\infty$.

**Assumption 4.2** (**LL objective**). Assume that the following conditions hold:

(i) For each $x \in X$, $f$ is convex *w.r.t.* LL variable $y$ on $Y$.

(ii) $f$ is continuously differentiable on an open set containing $X \times Y$ and is $L_f$-smooth on $X \times Y$.

**Assumption 4.3** (**LL constraints**). Assume that the following conditions hold:

(i) For each $x \in X$, $g$ is convex *w.r.t.* LL variable $y$ on $Y$.

(ii) $g(x, y)$ is $L_g$-Lipschitz continuous on $X \times Y$.

(iii) $g(x, y)$ is continuously differentiable on an open set containing $X \times Y$, $\nabla_x g(x, y)$ and $\nabla_y g(x, y)$ are $L_{g_1}$ and $L_{g_2}$-Lipschitz continuous on $X \times Y$, respectively.

Our assumptions solely require the first-order differentiability of the problem functions, avoiding (possibly) higher-order smoothness. The setting of this study substantially relaxes the existing requirement for second-order differentiability in constrained BiO literature. Moreover, we do not impose either the strong convexity on the LL objective $f(x, \cdot)$ or the fully convexity of the LL constraints $g(x, y)$.

### 4.2 Convergence results

The proof of non-asymptotic convergence for BiC-GAFFA primarily hinges on establishing the sufficient descent property of the merit function defined as follows:

$$V_k := \phi_{c_k}(x^k, y^k, z^k) + C_\theta \left\| \theta^k - \theta^*(x^k, y^k, z^k) \right\|^2,$$

where

$$\phi_{c_k}(x, y, z) := \frac{1}{c_k} \left( F(x, y) - \underline{F} \right) + \mathcal{G}_\gamma(x, y, z),$$

and $C_\theta := (L_f + rL_{g_1} + \frac{1}{\gamma_1} + L_g)^2$. With appropriately chosen step sizes $\alpha_k$ and $\eta_k$, the following inequality holds

$$V_{k+1} - V_k \leq -\frac{1}{4\alpha_k}\left\|w^{k+1} - w^k\right\|^2 - \frac{\eta_k C_\theta}{2\gamma_1}\|\theta^k - \theta^*(w^k)\|^2,$$

where $w^k := (x^k, y^k, z^k)$. See Lemma A.5 for a detailed description.

We define the residual function for problem (15) as follows

$$R(x, y, z) := \text{dist}\left(0, \nabla\psi_c(x, y, z) + \mathcal{N}_{X \times Y \times Z}(x, y, z)\right),$$

noting that $R(x, y, z) = 0$ if and only if $(x, y, z)$ is a stationary point to (15), i.e.,

$$0 \in \nabla\psi_c(x, y, z) + \mathcal{N}_{X \times Y \times Z}(x, y, z).$$

**Theorem 4.4.** *Under Assumptions 4.1, 4.2 and 4.3, assume that $X$, $Y$ are compact sets, $\gamma_1 > 0$, $\gamma_2 > 0$, $c > 0$ and $\eta_k \in (\underline{\eta}, 1/(L_f + rL_{g_2} + 1/\gamma_1))$ with $\underline{\eta} > 0$. Then there exists $c_\alpha > 0$ such that when $\alpha_k \in (\underline{\alpha}, c_\alpha)$ with $\underline{\alpha} > 0$, the sequence of $(x^k, y^k, z^k, \theta^k, \lambda^k)$ generated by Algorithm 1 satisfies*

$$\min_{0 \leq k \leq K} R(x^{k+1}, y^{k+1}, z^{k+1}) = O\left(\frac{1}{K^{1/2}}\right).$$

In subsequent analysis, we consider Algorithm 1 and the penalty approximation problem (15) with an iteratively increasing penalty parameter $c_k$, leading to the following convergence result based on the residual function $R_k(x, y, z) := \text{dist}\left(0, \nabla\psi_{c_k}(x, y, z) + \mathcal{N}_{X \times Y \times Z}(x, y, z)\right)$.

**Theorem 4.5.** *Under Assumptions 4.1, 4.2 and 4.3, assume that $X$, $Y$ are compact sets, $\gamma_1 > 0$, $\gamma_2 > 0$, $c_k = c(k+1)^\rho$ with $c > 0$, $\rho \in [0, 1/2)$ and $\eta_k \in (\underline{\eta}, 1/(L_f + rL_{g_2} + 1/\gamma_1))$ with $\underline{\eta} > 0$. Then there exists $c_\alpha > 0$ such that when $\alpha_k \in (\underline{\alpha}, c_\alpha)$ with $\underline{\alpha} > 0$, the sequence of $(x^k, y^k, z^k, \theta^k, \lambda^k)$ generated by Algorithm 1 satisfies*

$$\min_{0 \leq k \leq K} R_k(x^{k+1}, y^{k+1}, z^{k+1}) = O\left(\frac{1}{K^{(1-2\rho)/2}}\right).$$

*Furthermore, if $\rho > 0$ and $\psi_{c_k}(x^k, y^k, z^k)$ is uniformly bounded above, then the sequence of $(x^k, y^k, z^k)$ satisfies*

$$0 \leq \mathcal{G}_\gamma(x^K, y^K, z^K) = O\left(\frac{1}{K^\rho}\right).$$

*Remark* 4.6. The hypergradient norm is commonly employed as a stationary measure for BiO problems, when the LL problem is unconstrained and strongly convex. However, for constrained BiO problems, even if the LL objective is (strongly) convex *w.r.t.* the LL variable, the differentiability of variants of hypergradient, including optimistic and pessimistic ones, is not fully understood. To our best knowledge, no universally recognized stationary measure is known in this scenario. Different methods use various stationary measures. For instance, the KKT residual function is utilized in (Liu et al., 2023b; Lu, 2023). The residual functions in the above theorems originate from the corresponding penalized problems, aligning with several previous prior on first-order penalty methods (Liu et al., 2023a; Shen & Chen, 2023; Lu & Mei, 2024; Kwon et al., 2024).

## 5 EXTENSION TO BILEVEL OPTIMIZATION WITH MINIMAX LOWER-LEVEL PROBLEM

In this section, we explore the extension of our proposed gradient-based, single-loop, Hessian-free algorithm, originally designed for bilevel optimization problems with constrained lower-level problems, to bilevel optimization problems with minimax lower-level problem (Beck et al., 2023; Sato et al., 2021),

$$\min_{x \in X, y \in Y, z \in Z} F(x, y, z) \quad \text{s.t.} \quad (y, z) \in \mathcal{SP}(x), \tag{16}$$

where $\mathcal{SP}(x)$ denotes the set of saddle points for the convex-concave minimax problem,

$$\min_{y \in Y} \max_{z \in Z} f(x, y, z), \tag{17}$$

where $x \in \mathbb{R}^n$, $y \in \mathbb{R}^m$ and $z \in \mathbb{R}^p$, the sets $X$, $Y$ and $Z$ are closed convex sets in $\mathbb{R}^n$, $\mathbb{R}^m$ and $\mathbb{R}^p$, respectively. The UL objective $F : X \times Y \times Z \to \mathbb{R}$, and the LL objective $f : X \times Y \times Z \to \mathbb{R}$ are continuously differentiable with $f$ being convex in $y$ and concave in $z$. Building upon the idea applied in the development of the regularized gap function (3), we introduce the doubly regularized gap function for lower-level minimax problems, defined as:

$$\mathcal{G}_\gamma^{\text{saddle}}(x, y, z) := \max_{\theta \in Y, \lambda \in Z} \left\{ f(x, y, \lambda) - \frac{1}{2\gamma_2} \|\lambda - z\|^2 - f(x, \theta, z) - \frac{1}{2\gamma_1} \|\theta - y\|^2 \right\}. \quad (18)$$

By employing proof techniques analogous to those used in Lemma 2.1 or Theorem 3.3 from Von Heusinger & Kanzow (2009), we can derive similar results for the doubly regularized gap function $\mathcal{G}_\gamma^{\text{saddle}}(x, y, z)$.

**Lemma 5.1.** *Assume that $f(x, y, z)$ is convex in $y$ on $Y$ for any given $x \in X, z \in Z$ and concave in $z$ on $Z$ for any given $x \in X, y \in Y$. Let $\gamma_1, \gamma_2 > 0$, we have $\mathcal{G}_\gamma^{\text{saddle}}(x, y, z) \geq 0$ for any $(x, y, z) \in X \times Y \times Z$, and*

$$\mathcal{G}_\gamma^{\text{saddle}}(x, y, z) \leq 0,$$

*if and only if $(y, z) \in \mathcal{SP}(x)$.*

This characteristic of the newly introduced gap function enables the following equivalent single-level reformulation of the problem (16),

$$\min_{(x, y, z) \in X \times Y \times Z} F(x, y, z) \quad \text{s.t.} \quad \mathcal{G}_\gamma^{\text{saddle}}(x, y, z) \leq 0. \quad (19)$$

Using Lemma 5.1 and the proof techniques in Theorem 5.2, we can establish the equivalence between the reformulation (19) and the bilevel optimization problem (16).

**Theorem 5.2.** *Assume that $f(x, y, z)$ is convex in $y$ on $Y$ for any given $x \in X, z \in Z$ and concave in $z$ on $Z$ for any given $x \in X, y \in Y$. Let $\gamma_1, \gamma_2 > 0$, the reformulation (19) is equivalent to the bilevel optimization problem (16).*

Utilizing the reformulation (19) and the gradient formula of the gap function $\mathcal{G}_\gamma^{\text{saddle}}(x, y, z)$ provided in Lemma A.7 in the Appendix, analogous to BiC-GAFFA, we can propose a gradient-based, single-loop, Hessian-free algorithm for problem (16), as detailed in Algorithm 2.

---

**Algorithm 2** single-loop Hessian-free algorithm for bilevel optimization problems with minimax lower-level problem

---

**Input:** $(x^0, y^0, z^0) \in X \times Y \times Z$, $\theta^0 \in Y$, stepsizes $\alpha_k, \eta_k > 0$, proximal parameter $\gamma = (\gamma_1, \gamma_2)$, penalty parameter $c_k$;
**for** $k = 0$ **to** $K - 1$ **do**
- calculate $d_\theta^k$ and $d_\lambda^k$ as in (55);
- update
$$\theta^{k+1} = \text{Proj}_Y \left( \theta^k - \eta_k d_\theta^k \right), \quad \lambda^{k+1} = \text{Proj}_Z \left( \lambda^k - \eta_k d_\lambda^k \right);$$
- calculate $d_x^k, d_y^k, d_z^k$ as in (56);
- update
$$(x^{k+1}, y^{k+1}, z^{k+1}) = \text{Proj}_{X \times Y \times Z} \left( (x^k, y^k, z^k) - \alpha_k(d_x^k, d_y^k, d_z^k) \right).$$

**end for**

---

While the primary focus of this paper is the bilevel optimization with constrained lower-level problems, we defer the convergence analysis of this algorithm to future work.

## 6 NUMERICAL EXPERIMENTS

To validate both the theoretical and practical performance of our proposed algorithm (BiC-GAFFA), we conduct experiments on both synthetic tests and real-world applications. We compare BiC-GAFFA with various related algorithms on the synthetic tests and some real-world applications. Detailed information about the experiments can be found in Appendix A.2. Our code is available at https://github.com/SUSTech-Optimization/BiC-GAFFA.

## 6.1 SYNTHETIC EXPERIMENTS

Here we consider the following synthetic bilevel optimization problem:

$$
\begin{aligned}
\min_{\substack{\mathbf{x}\in\mathcal{X}, \\ (\mathbf{y}_1,\mathbf{y}_2)\in\mathcal{Y}}} \quad & (\mathbf{y}_1 - 2\cdot\mathbb{1}_n)^{\mathrm{T}}(\mathbf{x} - \mathbb{1}_n) + \|\mathbf{y}_2 + 3\cdot\mathbb{1}_n\|^2 \\
\text{s.t.} \quad & (\mathbf{y}_1,\mathbf{y}_2)\in\operatorname*{argmin}_{(\mathbf{y}_1,\mathbf{y}_2)\in\mathcal{Y}}\Big\{\tfrac{1}{2}\|\mathbf{y}_1\|^2 - \mathbf{x}^{\mathrm{T}}\mathbf{y}_1 + \mathbb{1}_n^{\mathrm{T}}\mathbf{y}_2 \text{ s.t. } \sum_{i=1}^n h(\mathbf{x}_i) + \mathbb{1}_n^{\mathrm{T}}\mathbf{y}_1 + \mathbb{1}_n^{\mathrm{T}}\mathbf{y}_2 = 0\Big\},
\end{aligned}
\tag{20}
$$

where $\mathcal{X} = \mathbb{R}^n$, $\mathcal{Y} = \mathbb{R}^n \times \mathbb{R}^n$, and $h(t) : \mathbb{R} \mapsto \mathbb{R}$ is defined as $h(t) = t^q$. We consider the cases $q = 1$ and $q = 3$. Note in both cases, the optimal solution is $\mathbf{x}^* = \mathbb{1}_n$, $\mathbf{y}_1^* = 2\cdot\mathbb{1}_n$, $\mathbf{y}_2^* = -3\cdot\mathbb{1}_n$, and $S(\mathbf{x}) = \{(\mathbf{x} + \mathbb{1}_n, \mathbf{y}_2)| \sum_{i=1}^n h(\mathbf{x}_i) + \mathbb{1}_n^{\mathrm{T}}(\mathbf{x} + \mathbb{1}_n) + \mathbb{1}_n^{\mathrm{T}}\mathbf{y}_2 = 0\}$, where $\mathbb{1}_n$ denotes the $n$-dimensional vector with all elements equal to 1.

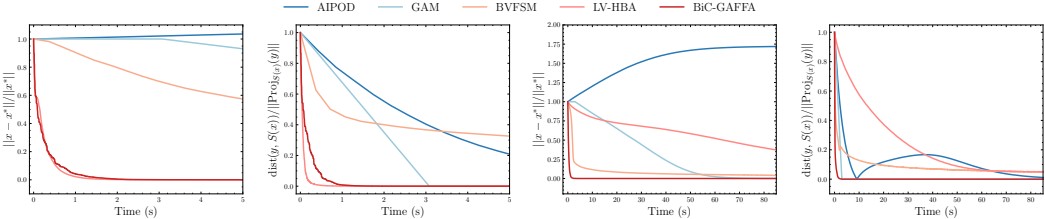

Figure 1: The first two pictures are results for the problem (20) with $q = 1, n = 1000$, and the third and fourth pictures are results for the problem (20) with $q = 3, n = 1000$.

We compare our algorithm (BiC-GAFFA) with AiPOD (Xiao et al., 2023), GAM (Xu & Zhu, 2023), BVFSM (Liu et al., 2023a), and LV-HBA (Yao et al., 2024) on the synthetic tests. Hyperparameters are collected in Appendix A.2.1. The comparison results are collected in Figure 1. From this, we can see that BiC-GAFFA converges fast and correctly in both cases, while AiPOD fails to optimize either of the problems, algorithm GAM and BVFSM run slowly on both cases, the algorithm LV-HBA converges well in the case of $q = 1$ but runs slowly in the case of $q = 3$ due to the high complexity of the projection step.

Table 1: Sensitivity analysis on problem (20) with $q = 3, n = 100$.

| $\gamma_1$ | $\gamma_2$ | $\alpha_k$ | $\eta_k$ | $p$ | Time (s) | $\gamma_1$ | $\gamma_2$ | $\alpha_k$ | $\eta_k$ | $p$ | Time (s) |
|---|---|---|---|---|---|---|---|---|---|---|---|
| 10 | 1.0 | 0.001 | 0.01 | 0.3 | 2.24 | 10 | 0.7 | 0.001 | 0.01 | 0.3 | 2.88 |
| 7 | 1.0 | 0.001 | 0.01 | 0.3 | 3.04 | 10 | 1.0 | 0.0001 | 0.001 | 0.3 | 22.74 |
| 5 | 1.0 | 0.001 | 0.01 | 0.3 | 2.47 | 10 | 1.0 | 0.0003 | 0.003 | 0.3 | 7.69 |
| 3 | 1.0 | 0.001 | 0.01 | 0.3 | 2.28 | 10 | 1.0 | 0.0005 | 0.005 | 0.3 | 4.64 |
| 1 | 1.0 | 0.001 | 0.01 | 0.3 | 2.66 | 10 | 1.0 | 0.0007 | 0.007 | 0.3 | 3.25 |
| 10 | 0.1 | 0.001 | 0.01 | 0.3 | 2.89 | 10 | 1.0 | 0.001 | 0.01 | 0.1 | 0.45 |
| 10 | 0.3 | 0.001 | 0.01 | 0.3 | 2.72 | 10 | 1.0 | 0.001 | 0.01 | 0.2 | 1.34 |
| 10 | 0.5 | 0.001 | 0.01 | 0.3 | 2.60 | 10 | 1.0 | 0.001 | 0.01 | 0.4 | 2.92 |
|  |  |  |  |  |  | 10 | 1.0 | 0.001 | 0.01 | 0.49 | 1.95 |

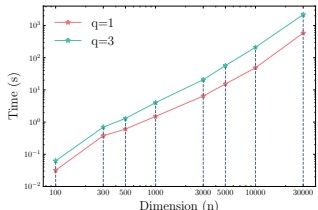

Figure 2: Time cost v.s. $n$.

**Sensitivity analysis.** We conduct a sensitivity analysis on the problem (20) to show the performance of BiC-GAFFA under different hyperparameters and problem sizes. We use the computational time when $\|\mathbf{x}_k - \mathbf{x}^*\|/\|\mathbf{x}^*\| < 0.01$ as a metric to measure the performance. The comparison results for different hyperparameters are presented in Table 1, which demonstrates the robustness of BiC-GAFFA. Figure 2 shows the results based on the problem's dimension, which is plotted on logarithmic scales for both axes, indicating that BiC-GAFFA scales well for large-scale problems. Detailed reports and additional information are provided in Appendix A.2.1.

Table 2: Results on the sparse group Lasso hyperparameter selection problem with nTr = 100, nVal = 100, nTest = 300.

| Method | Time(s) | Val Err | Test Err |
|---|---|---|---|
| Grid | $17.3 \pm 0.9$ | $35.9 \pm 7.2$ | $37.7 \pm 6.7$ |
| Random | $17.4 \pm 0.7$ | $33.6 \pm 6.7$ | $35.7 \pm 6.2$ |
| TPE | $16.9 \pm 0.7$ | $33.9 \pm 7.0$ | $36.0 \pm 5.6$ |
| IGJO | $21.2 \pm 2.2$ | $19.7 \pm 2.8$ | $25.6 \pm 4.4$ |
| VF-iDCA | $12.4 \pm 0.5$ | $14.6 \pm 2.6$ | $25.4 \pm 3.9$ |
| BiC-GAFFA | $21.4 \pm 0.7$ | $7.3 \pm 1.3$ | $22.3 \pm 3.0$ |

## 6.2 HYPERPARAMETER OPTIMIZATION

Hyperparameter optimization (HO) is an inherent bilevel optimization. According to Theorem 3.1 of Gao's work (Gao et al., 2022), we can turn the hyperparameter optimization problem of a statistical

learning model into an equivalent constrained BiO problem. Three HO problems are considered in this section. Detailed results and information are provided in Appendix A.2.2.

**Sparse group LASSO problem.** For the HO problem for the sparse group LASSO model, we compare BiC-GAFFA with some widely-used algorithms, including Grid Search, Random Search, TPE (Bergstra et al., 2013), IGJO (Feng & Simon, 2018), and VF-iDCA (Gao et al., 2022). Detailed settings are provided in Appendix A.2.2. From Table 2, we can see that BiC-GAFFA outperforms the other algorithms in terms of both validation and test errors.

**Support vector machine (SVM).** In this part, we apply BiC-GAFFA to a HO problem where the lower model (base learner) is a SVM and the upper model is a signed distance based loss. We compare BiC-GAFFA with GAM (Xu & Zhu, 2023) and LV-HBA and the results are shown in Figure 3. We can see that BiC-GAFFA converges faster than GAM and achieves a better performance.

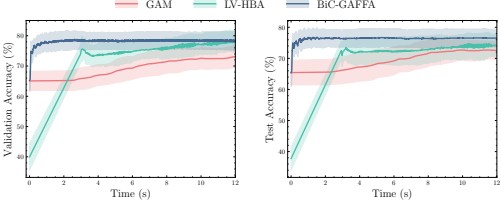
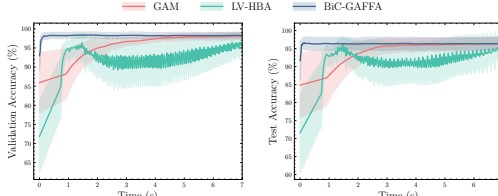

Figure 3: Results of the SVM problem on the diabetes dataset.

Figure 4: Results of the data hyper-cleaning problem on the breast-cancer dataset.

**Data hyper-cleaning.** Data hyper-cleaning (Franceschi et al., 2017) is to train a classifier based on data where part of their labels are corrupted with a probability $p_c$, to achieve this goal, we need to train a classifier that can recognize the wrong labeled data. Such a process can be modeled as a HO problem. Results are shown in Figure 4, from which we can see that BiC-GAFFA outperforms the other algorithms in terms of both validation and test accuracies.

## 6.3 GENERATIVE ADVERSARIAL NETWORKS

Generative adversarial networks (GAN) (Goodfellow et al., 2014) models are also investigated in our paper. It involves a hierarchical structure of optimization with two interacting components: the generator and the discriminator. We compared different training strategies for a distribution recovery problem. Detailed information is provided in Appendix A.2.3. The numerical results in Figure 5 and Table 3 show BiC-GAFFA can train such a generative model efficiently.

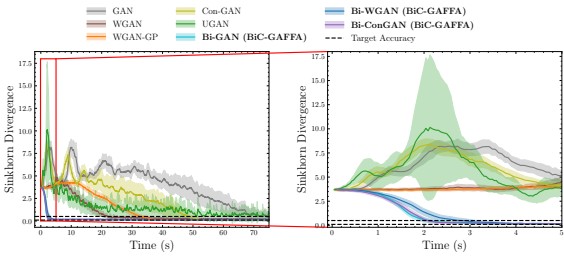

Figure 5: Earth mover's distance v.s Time.

Table 3: Time to meet the required accuracy, ($n*$) records the number of failures.

| Method | Time to reach target accuracy | |
| --- | --- | --- |
| | EM< 0.5 | EM< 0.1 |
| GAN | $60.0 \pm 13.0$ | $64.5 \pm 12.5$ |
| WGAN | $17.2 \pm 2.3$ | $32.1 \pm 6.0$ |
| WGAN-GP | $31.5 \pm 3.5$ | $37.7 \pm 6.2$ |
| Con-GAN | $36.4 \pm 11.5$ | $39.7 \pm 12.3$ |
| UGAN | $22.3 \pm 17.0$ (3*) | $24.1 \pm 18.2$ (4*) |
| **Bi-GAN (BiC-GAFFA)** | $2.0 \pm 1.0$ | $6.3 \pm 4.3$ |
| **Bi-WAN (BiC-GAFFA)** | $2.6 \pm 0.5$ | $29.4 \pm 69.0$ |
| **Bi-ConGAN (BiC-GAFFA)** | $2.1 \pm 0.2$ | $7.9 \pm 10.5$ |

## 7 DISCUSSION AND CONCLUSION

This work introduces BiC-GAFFA, a single-loop first-order algorithm tailored for a broader class of bilevel optimization problems, where the LL constraints may depend on the UL variables. A key technique employed here to address constrained LL problem is the newly introduced doubly regularized gap function, which serves as an optimality metric for the LL problems and allows for straightforward gradient evaluation. We also study the non-asymptotic performance guarantees of BiC-GAFFA. The experimental results validate the effectiveness of BiC-GAFFA.

ACKNOWLEDGMENTS

Authors listed in alphabetical order. This work is supported by National Natural Science Foundation of China (12222106, 12326605, 62331014, 12371305), Guangdong Basic and Applied Basic Research Foundation (No. 2022B1515020082).

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

# A APPENDIX

The appendix is organized as follows:

- We give a brief review of additional related work in Section A.1.
- Experimental details are provided in Section A.2.
- Proofs in Section 2 can be found in Section A.3.
- Proofs in Section 3 are presented in Section A.4.
- Proofs in Section 4 are given in Section A.5.
- Details regarding the extension to bilevel optimization with minimax lower-level problem are discussed in Section A.6.

## A.1 ADDITIONAL RELATED WORK

In the section, we give a brief review of additional related works that are directly related to ours.

**Gap functions.** Transforming a BiO problem into a single-level optimization problem is useful both theoretically and computationally. Gap functions are crucial in this process. Among them, the most commonly used one is the value function, originally proposed in (Outrata, 1990; Ye & Zhu, 1995). When the LL problem is unconstrained and its smooth objective is strongly convex *w.r.t.* the LL variable, the value function of the LL problem is smooth and has a closed-form gradient with the LL solution. Benefiting from this property, fully first-order gradient-based algorithms have been developed, see, *e.g.*, Liu et al. (2023a); Ye et al. (2022); Shen & Chen (2023); Kwon et al. (2023); Lu & Mei (2024). However, the value function is often nonsmooth when the LL problem is constrained or its objective is merely convex. As a result, the value function reformulation usually leads to a nonsmooth single-level optimization problem. Recently, the Moreau envelope-based gap functions have emerged, see, *e.g.*, Gao et al. (2023); Yao et al. (2024); Liu et al. (2024). Among these works, by utilizing the Moreau envelope-based reformulation for unconstrained LL problem, the very recent study (Liu et al., 2024) proposes a novel single-loop gradient-based algorithm for general BiO problems with nonconvex and potentially nonsmooth LL objective functions. In contrast, both Gao et al. (2023) and Yao et al. (2024) address constrained BiO problems. The key difference is that the Moreau envelope-based reformulation in Gao et al. (2023) is nonsmooth, while the reformulation in Yao et al. (2024) is smooth. The smoothness is achieved by leveraging a novel proximal Lagrangian value function to handle constrained LL problem. When the LL problem has (coupling) inequality constraints, it is important to note that all the reformulations in these works explicitly preserve these (coupling) inequality constraints. In contrast, this work introduces a novel reformulation that integrates the LL (coupling) inequality constraints into a doubly regularized gap function. Consequently, it achieves the minimal smooth constraint, with only one smooth constraint in the single-level reformulation. This provides a significant advantage in algorithm design because it removes the need for Euclidean projection onto the (coupled) LL constraint set. This expands the range of applications and significantly reduces computational costs.

**First-order algorithms.** Many applications involve optimization problems with thousands or millions of variables. First-order algorithms are popular because their storage and computational costs can be kept at a tolerable level. In bilevel optimization, the value function approach avoids recurrent second-order calculations like those involving the Hessian matrix in implicit gradient-based methods. This makes it particularly useful for developing efficient first-order, single-loop numerical algorithms, see, *e.g.*, Kwon et al. (2023); Chen et al. (2023), in cases where the LL problem is unconstrained and its smooth objective is strongly convex *w.r.t.* the LL variable. When the LL problem involves constraints, challenges arise. For instance, the value function is often nonsmooth and lacks an exact straightforward gradient evaluation. To address the nonsmooth issue, Liu et al. (2023a) introduces a sequential minimization algorithm framework using penalty and barrier functions. The recent study (Shen & Chen, 2023) provides a sufficient condition under which the value function is (Lipschitz) smooth and proposes first-order algorithms through the lens of the penalty method for bilevel problems with constraints that rely solely on the LL variable. Also, through the lens of the penalty method, a first-order method with complexity guarantees is developed in Lu & Mei (2024) using a novel minimax approach, and the recent study (Kwon et al., 2024) proposes a first-order stochastic bilevel optimization algorithm when the LL constraints depend only on the LL

variable. Other advances in first-order algorithms for bilevel optimization include: primal-dual algorithms (Sow et al., 2022); primal nonsmooth reformulation-based algorithm (Helou et al., 2023); and low-rank implicit gradient-based methods (Giovannelli et al., 2021).

## A.2 Details of experiments

All the experiments in this paper are performed on a computer with Intel(R) Core(TM) i9-9900K CPU @ 3.60GHz and 16.00 GB memory. Except for the synthetic experiments, the reported data for all the other experiments (including the sparse group LASSO problems, SVM, data hyperclean and GAN) are the statistical results after repeating each experiment 20 times. For further discussions on computational considerations and scalability, please refer to the notes provided in Appendix A.2.4.

### A.2.1 Synthetic problems

For all the experiments on this problem, we use the initial point $(\mathbf{x}_0, \mathbf{y}_0, \mathbf{y}_1) = (\mathbf{0}_n, \mathbf{0}_n, \mathbf{0}_n)$, where $\mathbf{0}_n$ represents the $n$-dimensional zero vector. For the synthetic experiments recorded in Figure 1, we use the following settings:

- For AIPOD, we set $S = 5, T = 2, \beta = 0.0001, \alpha = 0.0005$;
- For GAM, we solve the lower level problem with a convex solver Clarabel (Goulart & Chen, 2024) and set stepsize $\alpha = 0.01$;
- For BVFSM, we solve the subproblem (10) and (11) in (Liu et al., 2023a) with Clarabel, use the quadratic penalty function with $\theta_k = \mu_k = \rho_k = 1/k^{0.5}$ and stepsize $\alpha = 0.01$;
- For LV-HBA, we set $\gamma_1 = 10, \gamma_2 = 1, \alpha = 0.005, \beta = 0.002, \eta = 0.03$;
- For BiC-GAFFA, we set $\gamma_1 = 1, \gamma_2 = 0.1, \alpha = 0.001, \eta = 0.01, r = 1, \rho = 0.2$.

They are applied for both problems, i.e., for both $n = 1000, q = 1$ problem and $n = 1000, q = 3$ problem. Note that AIPOD requires the calculation of matrix inversion, which is computationally expensive, however, due to the special structure of the problem, the matrix inversion in each iteration returns the same result, so we only calculate it once and use the result in all iterations. For LV-HBA, the projection step is the most time-consuming part generally, it gets benefits from the special structure of the problem where $q = 1$ since the projection is efficient in that case, which is the reason why it performs well in the first case. However, for the problem where $q = 3$, the projection step is much more complex, which requires solving a nonlinear optimization problem, therefore, it performs poorly in the second case.

Table 4: Sensitivity analysis on the hyperparemeters for the problems with $n = 3$, where $q = 1$ in the left table and $q = 3$ in the right table.

| $\gamma_1$ | $\gamma_2$ | $\alpha_k$ | $\eta_k$ | $\rho$ | Time (s) | $\gamma_1$ | $\gamma_2$ | $\alpha_k$ | $\eta_k$ | $\rho$ | Time (s) |
|---|---|---|---|---|---|---|---|---|---|---|---|
| 1 | 1.0 | 0.001 | 0.01 | 0.3 | 2.96 | 1 | 1.0 | 0.001 | 0.01 | 0.3 | 2.66 |
| 3 | 1.0 | 0.001 | 0.01 | 0.3 | 2.84 | 3 | 1.0 | 0.001 | 0.01 | 0.3 | 2.28 |
| 5 | 1.0 | 0.001 | 0.01 | 0.3 | 2.95 | 5 | 1.0 | 0.001 | 0.01 | 0.3 | 2.47 |
| 7 | 1.0 | 0.001 | 0.01 | 0.3 | 2.63 | 7 | 1.0 | 0.001 | 0.01 | 0.3 | 3.04 |
| 10 | 0.1 | 0.001 | 0.01 | 0.3 | 3.02 | 10 | 0.1 | 0.001 | 0.01 | 0.3 | 2.89 |
| 10 | 0.3 | 0.001 | 0.01 | 0.3 | 3.14 | 10 | 0.3 | 0.001 | 0.01 | 0.3 | 2.72 |
| 10 | 0.5 | 0.001 | 0.01 | 0.3 | 2.98 | 10 | 0.5 | 0.001 | 0.01 | 0.3 | 2.60 |
| 10 | 0.7 | 0.001 | 0.01 | 0.3 | 3.09 | 10 | 0.7 | 0.001 | 0.01 | 0.3 | 2.88 |
| 10 | 1.0 | 0.0005 | 0.005 | 0.3 | 4.50 | 10 | 1.0 | 0.0001 | 0.001 | 0.3 | 21.98 |
| 10 | 1.0 | 0.0007 | 0.007 | 0.3 | 3.68 | 10 | 1.0 | 0.0003 | 0.003 | 0.3 | 7.28 |
| 10 | 1.0 | 0.003 | 0.03 | 0.3 | 0.83 | 10 | 1.0 | 0.0005 | 0.005 | 0.3 | 4.79 |
| 10 | 1.0 | 0.005 | 0.05 | 0.3 | 0.42 | 10 | 1.0 | 0.0007 | 0.007 | 0.3 | 3.30 |
| 10 | 1.0 | 0.001 | 0.01 | 0.1 | 0.59 | 10 | 1.0 | 0.001 | 0.01 | 0.1 | 0.45 |
| 10 | 1.0 | 0.001 | 0.01 | 0.2 | 1.76 | 10 | 1.0 | 0.001 | 0.01 | 0.2 | 1.34 |
| 10 | 1.0 | 0.001 | 0.01 | 0.4 | 2.66 | 10 | 1.0 | 0.001 | 0.01 | 0.4 | 2.92 |
| 10 | 1.0 | 0.001 | 0.01 | 0.49 | 2.00 | 10 | 1.0 | 0.001 | 0.01 | 0.49 | 2.04 |
| 10 | 1.0 | 0.001 | 0.01 | 0.3 | 2.20 | 10 | 1.0 | 0.001 | 0.01 | 0.3 | 2.51 |

To see how parameters impact the performance of the algorithms, we conduct a series of sensitivity analyses, the results are collected in Table 5. Besides, we also conduct experiments to investigate the

relationship between the problem's dimension and the time cost, the results are shown in Figure 2. For these expriments, we set $\gamma_1 = 10$, $\gamma_2 = 1$, $r = 1$, $\rho = 0.3$, and choose $(\alpha_k, \eta_k) = (20/n, 2/n)$ for problems with $q = 1$, $(\alpha_k, \eta_k) = (10/n, 1/n)$ for problems with $q = 3$. Such a choice is quite rough, but it is sufficient to demonstrate the scalability of BiC-GAFFA.

Table 5: Sensitivity analysis on the hyperparameters for the synthetic experiments.

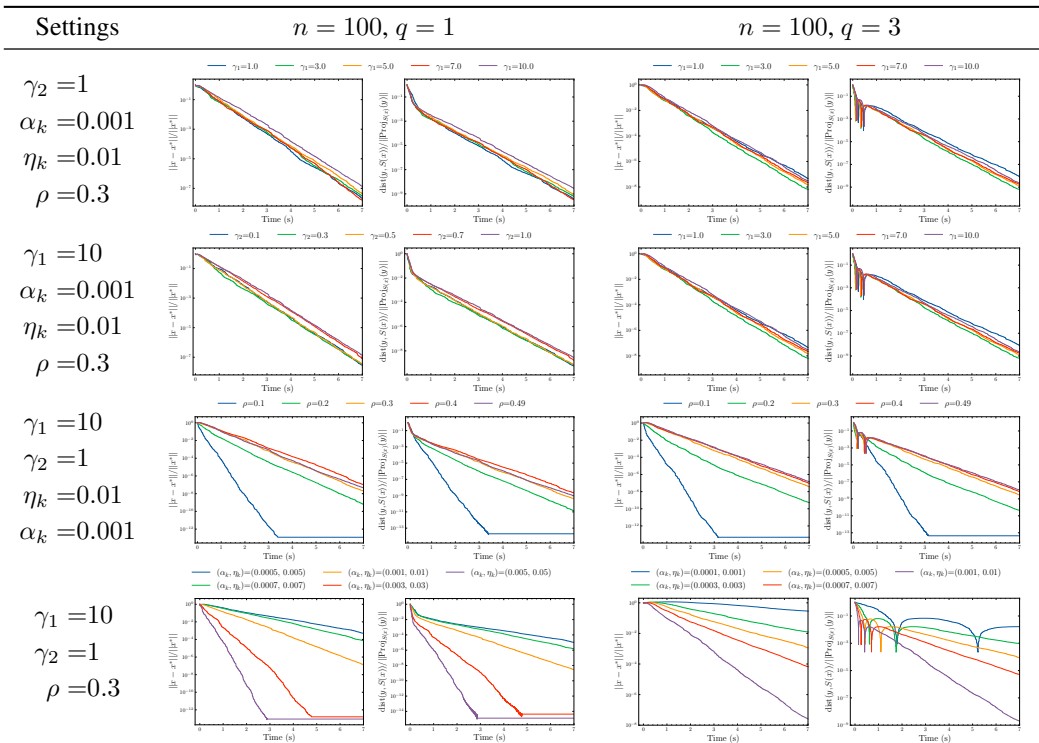

### A.2.2 HYPERPARAMETER OPTIMIZATION

Hyperparameter optimization is an inherent bilevel optimization. According to Theorem 3.1 of Gao's work (Gao et al., 2022), we can turn the hyperparameter optimization problem of a statistical learning model which consists of data fitting $\mathcal{L}_{\text{data}}(\mathbf{x})$ and convex regularization terms $P_i(\mathbf{x})$ to the following constrained bilevel optimization problem.

$$\min_{\mathbf{x} \in \mathbb{R}^n, \mathbf{r} \in \mathbb{R}^J} \mathcal{L}_{\text{val}}(\mathbf{x}) \ \text{s.t.} \ \mathbf{x} \in \underset{\mathbf{x} \in \mathbb{R}^n}{\arg\min} \left\{ \mathcal{L}_{\text{tr}}(\mathbf{x}) \ \text{s.t.} \ P_i(\mathbf{x}) \leq r_i, \ i = 1, \cdots, J \right\}.$$

**Sparse Group LASSO Problem.** The sparse group LASSO problem (Simon et al., 2013) is an advanced statistical learning model, which aims to find the grouped structure of the predictors and select the relevant ones. It is critical to select the weights of the regularization terms. The direct form of the problem is as follows:

$$\min_{\boldsymbol{\beta} \in \mathbb{R}^p, \boldsymbol{\lambda} \in \mathbb{R}^{M+1}_+} \frac{1}{2} \sum_{i \in I_{\text{val}}} |b_i - \boldsymbol{\beta}^{\mathrm{T}} \mathbf{a}_i|^2$$

$$\text{s.t.} \quad \boldsymbol{\beta} \in \underset{\hat{\boldsymbol{\beta}} \in \mathbb{R}^p}{\arg\min} \left\{ \frac{1}{2} \sum_{i \in I_{\text{tr}}} |b_i - \hat{\boldsymbol{\beta}}^{\mathrm{T}} \mathbf{a}_i|^2 + \sum_{m=1}^{M} \lambda_m \|\hat{\boldsymbol{\beta}}^{(m)}\|_2 + \lambda_{M+1} \|\hat{\boldsymbol{\beta}}\|_1 \right\}. \tag{21}$$

By decoupling the data fitting term and regularization terms, we can reformulate the problem to the following bilevel optimization problem:

$$
\begin{aligned}
\min_{\boldsymbol{\beta} \in \mathbb{R}^p, \mathbf{r} \in \mathbb{R}_+^{M+1}} \quad & \frac{1}{2} \sum_{i \in I_{\text{val}}} |b_i - \boldsymbol{\beta}^{\mathrm{T}} \mathbf{a}_i|^2 \\
\text{s.t.} \quad & \boldsymbol{\beta} \in \underset{\hat{\boldsymbol{\beta}} \in \mathbb{R}^p}{\arg\min} \frac{1}{2} \sum_{i \in I_{\text{tr}}} |b_i - \hat{\boldsymbol{\beta}}^{\mathrm{T}} \mathbf{a}_i|^2 \\
& \text{s.t.} \quad \|\hat{\boldsymbol{\beta}}^{(m)}\|_2 \leq r_m, m = 1, \ldots, M, \quad \|\hat{\boldsymbol{\beta}}\|_1 \leq r_{M+1}.
\end{aligned}
\tag{22}
$$

From our practice, we find that the proposed BiC-GAFFA algorithm works better on the squared two norms rather than two norms directly, so by introducing $\mathbf{u} \in \mathbb{R}_+^{M+1}$ such that $u_m = r_m^2$ for $m = 1, \ldots, M$, and $u_{M+1} = r_{M+1}$, we can reformulate the problem to the following bilevel optimization problem:

$$
\begin{aligned}
\min_{\boldsymbol{\beta} \in \mathbb{R}^p, \mathbf{u} \in \mathbb{R}_+^{M+1}} \quad & \frac{1}{2} \sum_{i \in I_{\text{val}}} |b_i - \boldsymbol{\beta}^{\mathrm{T}} \mathbf{a}_i|^2 \\
\text{s.t.} \quad & \boldsymbol{\beta} \in \underset{\hat{\boldsymbol{\beta}} \in \mathbb{R}^p}{\arg\min} \frac{1}{2} \sum_{i \in I_{\text{tr}}} |b_i - \hat{\boldsymbol{\beta}}^{\mathrm{T}} \mathbf{a}_i|^2 \\
& \text{s.t.} \quad \|\hat{\boldsymbol{\beta}}^{(m)}\|_2^2 \leq u_m, m = 1, \ldots, M, \quad \|\hat{\boldsymbol{\beta}}\|_1 \leq u_{M+1}.
\end{aligned}
\tag{23}
$$

Grid Search, Random Search, Bayesian Optimization and VF-iDCA are applied to the model (22), IGJO is applied to the model (21), and BiC-GAFFA is applied to the model (23). The experimental results are collected in Table 6.

Data used in these experiments are generated by the following procedures: we random smapled $\mathbf{a}_i \in \mathbb{R}^p$ from the standard normal distribution, and set the true weights $\boldsymbol{\beta}$ to be a grouped sparse vector, specifically, it was defined as $\boldsymbol{\beta} = [\boldsymbol{\beta}^{(1)}, \boldsymbol{\beta}^{(2)}, \ldots, \boldsymbol{\beta}^{(5)}]$, where the first 5 entries of $\boldsymbol{\beta}^{(i)}$ are 1, 2, 3, 4, 5, and the rest entries are all zeros. The responses are defined as $b_i = \boldsymbol{\beta}^T \mathbf{a}_i + \sigma \epsilon_i$, where the $\epsilon_i$ are generated from the standard normal distribution, and $\sigma$ is chosen such that the signal-to-noise ratio is 3. The training set, validation set, and test set are split randomly, and the size of the training set is denoted as nTr, the size of the validation set is nVal, and the size of the test set is nTest. For all the experiments in this part, $p = 150$, $M = 30$.

The detailed settings of each algorithm are provided as follows:

- For Grid Search, we set the grid size to be 20, the range of $r_m$ is $[1, 10]$ for $m = 1, \ldots, M$, and the range of $r_{M+1}$ is $[1, 100]$;

- For Random Search, we set the number of iterations to be 400, the range of $r_m$ is $[0, 10]$ for $m = 1, \ldots, M$, and the range of $r_{M+1}$ is $[0, 100]$, uniformly sampled method is adopted;

- For TPE, we set the number of iterations to be 400, and use uniform distribution for each hyperparameter, the range of $r_m$ is $[0, 10]$ for $m = 1, \ldots, M$, and the range of $r_{M+1}$ is $[0, 100]$;

- For IGJO, we set the number of iterations to be 100 for each hyperparameter, and the initial guess is $0.1 \times \mathbb{1}_{M+1}$;

- For VF-iDCA, we set the number of iterations to 50 for each hyperparameter. As to the initial guess, we firstly solve the lower level problem of (21) with hyperparameters $0.1 \times \mathbb{1}_{M+1}$, denote that $r_m = \|\hat{\boldsymbol{\beta}}^{(m)}\|_2$ for $m = 1, \ldots, M$, $r_{M+1} = \|\hat{\boldsymbol{\beta}}\|_1$, then take the $\mathbf{r}$ as the initial guess for the bilevel problem of (22);

- For BiC-GAFFA, we set the number of iterations to be 30000 for each hyperparameter. As to the initial guess, we firstly solve the lower level problem of (21) with hyperparameters $0.1 \times \mathbb{1}_{M+1}$, denote that $u_m = \|\hat{\boldsymbol{\beta}}^{(m)}\|_2^2$ for $m = 1, \ldots, M$, $u_{M+1} = \|\hat{\boldsymbol{\beta}}\|_1$, then take the $\mathbf{u}$ as the initial guess for the bilevel problem of (23). For this problem, we always take $\gamma_1 = 10$, $\gamma_2 = 1$, $\eta_k = 0.1$, $\alpha_k = 0.01$, $r = 0.5$, and $\rho = 0.3$.

From Table 6, we can see that BiC-GAFFA outperforms other algorithms in terms of the validation error and test error, and when the number of data is increased, the time cost of BiC-GAFFA does not increase significantly, which indicates that BiC-GAFFA is scalable to large-scale problems.

Table 6: Results on the sparse group Lasso hyperparameter selection problem.

| Method | nTr = 100, nVal = 100, nTest = 300 | | | nTr = 300, nVal = 300, nTest = 300 | | |
|--------|----------|----------|----------|----------|----------|----------|
| | Time (s) | Val Err | Test Err | Time (s) | Val Err | Test Err |
| Grid | $17.3 \pm 0.9$ | $35.9 \pm 7.2$ | $37.7 \pm 6.7$ | $78.7 \pm 1.9$ | $18.9 \pm 2.3$ | $19.8 \pm 1.8$ |
| Random | $17.4 \pm 0.7$ | $33.6 \pm 6.7$ | $35.7 \pm 6.2$ | $78.6 \pm 2.5$ | $18.7 \pm 2.4$ | $19.5 \pm 1.9$ |
| TPE | $16.9 \pm 0.7$ | $33.9 \pm 7.0$ | $36.0 \pm 5.6$ | $74.7 \pm 2.2$ | $18.9 \pm 2.3$ | $19.8 \pm 1.9$ |
| IGJO | $21.2 \pm 2.2$ | $19.7 \pm 2.8$ | $25.6 \pm 4.4$ | $49.9 \pm 2.6$ | $16.5 \pm 2.5$ | $18.1 \pm 1.4$ |
| VF-iDCA | $12.4 \pm 0.5$ | $14.6 \pm 2.6$ | $25.4 \pm 3.9$ | $40.7 \pm 1.7$ | $14.9 \pm 2.1$ | $17.2 \pm 1.3$ |
| BiC-GAFFA | $21.4 \pm 0.7$ | $7.3 \pm 1.3$ | $22.3 \pm 3.0$ | $22.0 \pm 1.0$ | $12.8 \pm 1.4$ | $17.1 \pm 1.3$ |

**Support Vector Machine.** The mathematical model can be written as follows:

$$\min_{\mathbf{c} \in \mathbb{R}^{N_{\mathrm{tr}}}, \mathbf{w} \in \mathbb{R}^p, b \in \mathbb{R}, \boldsymbol{\xi} \in \mathbb{R}^{N_{\mathrm{tr}}}} \mathcal{L}_{\mathcal{D}_{\mathrm{val}}}(\mathbf{w}, b),$$

$$\text{s.t.} \quad (\mathbf{w}, b, \boldsymbol{\xi}) \in \operatorname*{argmin}_{\mathbf{w} \in \mathbb{R}^p, b \in \mathbb{R}, \boldsymbol{\xi} \in \mathbb{R}^{N_{\mathrm{tr}}}} \frac{1}{2} \|\mathbf{w}\|^2 + \frac{1}{2} \sum_{i=1}^{N_{tr}} e^{c_i} \xi_i^2$$

$$\text{s.t.} \quad y_i(\mathbf{w}^{\mathrm{T}} \mathbf{x}_i + b) \geq 1 - \xi_i, \quad i = 1, \dots, N_{\mathrm{tr}}.$$

Here we use $(\mathcal{D}_{\mathrm{tr}}, \mathcal{D}_{\mathrm{val}})$ to denote the split of the training set and validation set of data, and $N_{\mathrm{tr}}$ to denote the number of data in the training set. The upper-level objective function is defined in the following way:

$$\mathcal{L}_{\mathcal{D}_{\mathrm{val}}}(\mathbf{w}, b) := \frac{1}{|\mathcal{D}_{\mathrm{val}}|} \sum_{(\mathbf{x}_i, y_i) \in \mathcal{D}_{\mathrm{val}}} \mathrm{Sigmoid} \left( -\frac{y_i(\mathbf{w}^{\mathrm{T}} \mathbf{x}_i + b)}{\|\mathbf{w}\|^2} \right).$$

The function inner the sigmoid function gives an opposite of a signed distance between point $(\mathbf{x}_i, y_i)$ and the decision hyperplane $\{\mathbf{x} \in \mathbb{R}^p | \mathbf{w}^{\mathrm{T}} \mathbf{x} + b = 0\}$. Specifically, the inner part is positive when the sign of the prediction $\mathbf{w}^{\mathrm{T}} \mathbf{x}_i + b$ is different with its label $y_i$, and negative otherwise. And the sigmoid function converts the distance values to some probability value. Such a composition makes the objective function to be a smooth approximation of the validation accuracy.

The dataset we used in this experiment is the scaled diabetes dataset provided in the repository (Chang & Lin, 2011), which contains 768 data points and 8 features. In each experiment, the training set, validation set, and test set are split randomly, and the size of the training set is 400, and the size of the validation set is 150, the rest part is the test set.

The initial values of $\mathbf{c}$ in this problem are sampled from a uniform distribution on $[-6, -5]$, the initial values for other parameters are solutions of the lower level problem with hyperparameters $\mathbf{c}$. The hyperparameters for all the three algorithms are:

- For GAM, we keep the same setting as the original paper (Xu & Zhu, 2023) did, the maximal iteration number is set to be 80.

- For LV-HBA, we use $\alpha = 0.001$, $\eta = 0.001$, $\gamma_1 = 0.1$, $\gamma_2 = 0.1$, the maixmal iteration number is set to be 400.

- For BiC-GAFFA, we use $\gamma_1 = 10$, $\gamma_2 = 0.01$, $\eta_k = 0.01$, $r = 10$, $\alpha_k = 0.001$, $\rho = 0.3$, the maximal iteration number is set to be 5000.

**Data Hyperclean.** For this problem, we use the same model as the one in the SVM experiments, but we change the dataset to the scaled breast-cancer dataset provided in the repository (Chang & Lin, 2011), which contains 683 data points and 10 features. In each experiment, the training set, validation set, and test set are split randomly, and the size of the training set is 400, and the size of the validation set is 180, the rest part is the test set. For the training dataset, we change the label with probability $p_c = 40\%$, and the validation set and test set are kept unchanged. This means the data used in lower training is not reliable, the researcher need to find out which data are more reliable and give them higher weights while giving the unreliable data lower weights. Such a weighting process can be viewed as a data clean procedure. Here we do such a procedure by regarding the weights as hyperparameters and evaluating their effects on the validation set, such a process is regarded as data

hyper cleaning. Therefore, such a problem is quite similar to the aforementioned SVM, the main difference occurs in the training data.

The hyperparameters for all the three algorithms are:

- For GAM, we keep the same setting as the original paper (Xu & Zhu, 2023) did, the maximal iteration number is set to be 30. The initial values of $\mathbf{c}$ are chosen from a uniformed distribution $(1., 2.)$.

- For LV-HBA, we use $\alpha = 0.001$, $\eta = 0.001$, $\gamma_1 = 0.1$, $\gamma_2 = 0.1$, the maximal iteration number is set to be 400. The initial values of $\mathbf{c}$ are chosen from a uniformed distribution $(-5, -4)$;

- For BiC-GAFFA, we use $\gamma_1 = 10$, $\gamma_2 = 0.01$, $\eta_k = 0.1$, $\alpha_k = 0.001$, $r = 10$, $\rho = 0.3$, the maximal iteration number is set to be 2000. The initial values of $\mathbf{c}$ are choosen from a unifromed distribution $(-5, -4)$.

### A.2.3 GENERATIVE ADVERSARIAL NETWORK

We also apply our algorithm to the GAN, which is a popular model in the field of deep learning. It can be written as the following bilevel optimization problem:

$$\min_{G,D} \mathcal{L}_{\text{gen}}(G, D) \quad \text{s.t.} \quad D \in \text{argmin}_{D \in \mathcal{D}} \mathcal{L}_{\text{det}}(G, D)$$

The main idea of GAN is to find a network that learns the distribution of the given training data. This paper investigates the fitting of two-dimensional distributions (i.e., 8-Gaussians model and 25-Gaussians model, refer to Gulrajani et al. (2017)) using GANs. For such simple distributions, we can approximate the earth mover's distance (EMD) between the generated distribution and the true distribution by calculating the Sinkhorn distance (Cuturi, 2013). This allows us to observe the optimization progress of the network during the iteration process, and make comparisons between different strategies.

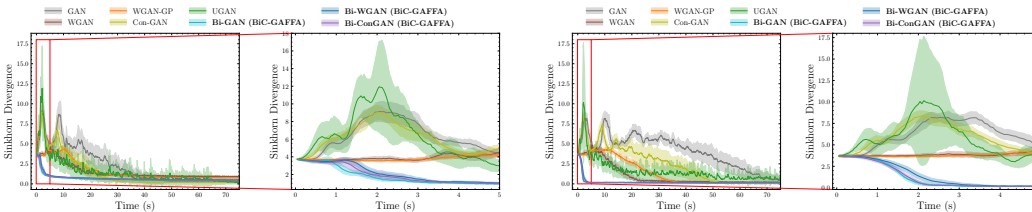

Figure 6: How EMD changes w.r.t time for different GAN models. The true distribution in the left picture is the 8 Gaussians mixture model, and the true distribution in the right picture is the 25 Gaussians mixture model.

For all the 8 GANs we make a comparison here, we use similar structures and similar parameters, though the loss functions exhibit slight variations. The generator and discriminator are both three-layer neural networks with 128 hidden units, and the activation function except the last layer is ReLU. The output size of the generator is 256. For the WGAN and WGAN-GP, the last layer of the discriminator is the linear layer while other models have a sigmoid activation function after the linear layer in their discriminator. The batch size is 512, and the number of iterations is 2701. According to the practical advice in Gulrajani et al. (2017), we train the discriminator 5 steps and then the generator 1 step in 1 loop. We use Adam with $b1 = 0.5, b2 = 0.999$ for all the learnable parameters in all the models. The specific objectives and additional parameter details are listed below:

- GAN (Goodfellow et al., 2014):

$$\mathcal{L}_{\text{gen}}(G, D) = \mathbb{E}_{\mathbf{z} \sim \mathbb{N}}[\text{BCELoss}(D(G(\mathbf{z})), \mathbb{1}_n)],$$

$$\mathcal{L}_{\text{det}}(G, D) = \frac{1}{2}\mathbb{E}_{\mathbf{x} \sim \mathbb{P}_r}[\text{BCELoss}(D(\mathbf{x}), \mathbb{1}_n)] + \frac{1}{2}\mathbb{E}_{\mathbf{z} \sim \mathbb{N}}[\text{BCELoss}(D(G(\mathbf{z})), \mathbf{0}_n)].$$

We set learning rate of generator as $10^{-3}$, learning rate of discriminator as $10^{-4}$.

- WGAN (Arjovsky et al., 2017):

$$\mathcal{L}_{\text{gen}}(G, D) = - \mathbb{E}_{\mathbf{z} \sim \mathbb{N}}[D(G(\mathbf{z})],$$
$$\mathcal{L}_{\text{det}}(G, D) = - \mathbb{E}_{\mathbf{x} \sim \mathbb{P}_r}[D(\mathbf{x})] + \mathbb{E}_{\mathbf{z} \sim \mathbb{N}}[D(G(\mathbf{z})],$$

with a box constraint that $\|D\|_{\infty} \leq c_{\text{clip}}$. We set learning rate of generator as $10^{-3}$, learning rate of discriminator as $10^{-4}$, the clip value as 0.001;

- WGAN-GP (Gulrajani et al., 2017):

$$\mathcal{L}_{\text{gen}}(G, D) = - \mathbb{E}_{\mathbf{z} \sim \mathbb{N}}[D(G(\mathbf{z})],$$
$$\mathcal{L}_{\text{det}}(G, D) = - \mathbb{E}_{\mathbf{x} \sim \mathbb{P}_r}[D(\mathbf{x})] + \mathbb{E}_{\mathbf{z} \sim \mathbb{N}}[D(G(\mathbf{z})] + \lambda_{\text{gp}} \mathbb{E}_{\hat{\mathbf{x}} \sim \mathbb{P}_{\hat{\mathbf{x}}}}(\|\nabla D(\hat{\mathbf{x}})\|_2 - 1)^2,$$

where $\lambda_{\text{gp}}$ is the parameter of gradient penalty, and $\mathbb{P}_{\hat{\mathbf{x}}}$ is defined by sampling uniformly along straight lines between pairs of points sampled from the data distribution $\mathbb{P}_r$ and the generator distribution $\mathbb{P}_g := \{G(\mathbf{z})|\mathbf{z} \in \mathbb{N}\}$. We set learning rate of generator as $10^{-4}$, learning rate of discriminator as $10^{-4}$, lambda of gradient penalty as 0.1;

- Con-GAN (Chao et al., 2021):

$$\mathcal{L}_{\text{gen}}(G, D) = \mathbb{E}_{\mathbf{z} \sim \mathbb{N}}[\text{BCELoss}(D(G(\mathbf{z})), \mathbb{1}_n)],$$
$$\mathcal{L}_{\text{det}}(G, D) = \mathbb{E}_{\mathbf{x} \sim \mathbb{P}_r}[\text{BCELoss}(D(\mathbf{x}), \mathbb{1}_n)] + \mathbb{E}_{\mathbf{z} \sim \mathbb{N}}[\text{BCELoss}(D(G(\mathbf{z})), _n)] + \lambda h(G, D),$$

where

$$h(G, D) := \mathbb{E}_{\{\mathbf{x} \sim \mathbb{P}_r, \mathbf{z} \sim \mathbb{N}\}}[\log(D(\mathbf{x})) - \log D(G(\mathbf{z}))]^2.$$

We set learning rate of generator as $10^{-3}$, learning rate of discriminator as $10^{-4}$, lambda of constant penalty as 0.3;

- UGAN (Metz et al., 2016):

$$\mathcal{L}_{\text{gen}}(G, D) = \mathbb{E}_{\mathbf{z} \sim \mathbb{N}}[\text{BCELoss}(D(G(\mathbf{z})), \mathbb{1}_n)],$$
$$\mathcal{L}_{\text{det}}(G, D) = \frac{1}{2}\mathbb{E}_{\mathbf{x} \sim \mathbb{P}_r}[\text{BCELoss}(D(\mathbf{x}), \mathbb{1}_n)] + \frac{1}{2}\mathbb{E}_{\mathbf{z} \sim \mathbb{N}}[\text{BCELoss}(D(G(\mathbf{z})), _n)].$$

We set the learning rate of the generator as $10^{-3}$, the learning rate of the discriminator as $10^{-4}$, the unrolled step as 5 and let it update the discriminator just 1 step in 1 loop;

- Bi-GAN (BiC-GAFFA):

$$\mathcal{L}_{\text{gen}}(G, D) = \mathbb{E}_{\mathbf{z} \sim \mathbb{N}}[\text{BCELoss}(D(G(\mathbf{z})), \mathbb{1}_n)],$$
$$\mathcal{L}_{\text{det}}(G, D) = \frac{1}{2}\mathbb{E}_{\mathbf{x} \sim \mathbb{P}_r}[\text{BCELoss}(D(\mathbf{x}), \mathbb{1}_n)] + \frac{1}{2}\mathbb{E}_{\mathbf{z} \sim \mathbb{N}}[\text{BCELoss}(D(G(\mathbf{z})), _n)].$$

We set the learning rate of the generator as $10^{-3}$, the learning rate of the discriminator as $10^{-4}$, $\rho = 0.1$, the learning rate of the auxiliary generator as $10^{-3}$, the upper bound for the auxiliary dual variable $(r)$ as 0.1;

- Bi-WGAN (BiC-GAFFA):

$$\mathcal{L}_{\text{gen}}(G, D) = - \mathbb{E}_{\mathbf{z} \sim \mathbb{N}}[D(G(\mathbf{z})],$$
$$\mathcal{L}_{\text{det}}(G, D) = - \mathbb{E}_{\mathbf{x} \sim \mathbb{P}_r}[D(\mathbf{x})] + \mathbb{E}_{\mathbf{z} \sim \mathbb{N}}[D(G(\mathbf{z})],$$

with constraint

$$\max_{\hat{\mathbf{x}} \in \mathbb{P}_{\hat{\mathbf{x}}}} \|\nabla D(\hat{\mathbf{x}})\|_2 \leq 1,$$

where $\mathbb{P}_{\hat{\mathbf{x}}}$ is defined by sampling uniformly along straight lines between pairs of points sampled from the data distribution $\mathbb{P}_r$ and the generator distribution $\mathbb{P}_g := \{G(\mathbf{z})|\mathbf{z} \in \mathbb{N}\}$. We set the learning rate of the generator as $10^{-3}$, the learning rate of the discriminator as $10^{-4}$, $r = 0.1$, $\rho = 0.1$, the learning rate of the auxiliary generator as $10^{-3}$, the learning rate of the auxiliary dual variable as $10^{-4}$, the upper bound for the auxiliary dual variable $(r)$ as 0.1. Note here we turn the gradient penalty introduced in Gulrajani et al. (2017) to one constraint by taking a maximum over the sample points, and by our practice, the original GAN structure seems more stable with this optimization method, further study is still required;

- Bi-ConGAN (BiC-GAFFA):

$$\mathcal{L}_{\text{gen}}(G, D) = \mathbb{E}_{\mathbf{z} \sim \mathbb{N}}[\text{BCELoss}(D(G(\mathbf{z})), \mathbb{1}_n)],$$
$$\mathcal{L}_{\text{det}}(G, D) = \mathbb{E}_{\mathbf{x} \sim \mathbb{P}_r}[\text{BCELoss}(D(\mathbf{x}), \mathbb{1}_n)] + \mathbb{E}_{\mathbf{z} \sim \mathbb{N}}[\text{BCELoss}(D(G(\mathbf{z})), _n)],$$

with constraint

$$h(G, D) := \mathbb{E}_{\{\mathbf{x} \sim \mathbb{P}_r, \mathbf{z} \sim \mathbb{N}\}}[\log(D(\mathbf{x})) - \log D(G(\mathbf{z}))]^2 \le \epsilon.$$

We set the learning rate of the generator as $10^{-3}$, the learning rate of the discriminator as $10^{-4}$, $\rho = 0.1$, the learning rate of the auxiliary generator as $10^{-3}$, the learning rate of the auxiliary dual variable ($r$) as $10^{-4}$, the upper bound for the auxiliary dual variable as 0.1. We realize the constraint proposed in Chao et al. (2021) with $\varepsilon = 0.1$.

Different analyses and assumptions lead to various regularization requirements for these models. When constrained optimizers are unavailable, such a regularization method can only be achieved by penalizing them to the objective (Gulrajani et al., 2017; Chao et al., 2021), potentially compromising the interpretability of the model and complicating analysis. Our algorithms mitigate these issues without significant computational overhead. The effectiveness of our algorithms is demonstrated in the numerical results presented in Figures 6.

### A.2.4 ADDITIONAL NOTES ON NUMERICAL EXPERIMENTS

**Computational and Memory Considerations**

We acknowledge that the introduction of additional sequences, such as $\theta_k$ and $\lambda_k$, does contribute to increased memory usage, which can present challenges for large-scale applications. Addressing this limitation and exploring ways to reduce the storage requirements is an area we plan to investigate in future work.

In terms of computational complexity, while our algorithm introduces additional computation, our empirical results suggest that these additional calculations contribute to more effective model optimization. This added complexity, in many cases, proves competitive and can lead to improved convergence and solution quality. Moreover, from the perspective of numerical implementation, our method offers notable advantages over some traditional bilevel optimization algorithms. Unlike conventional approaches, our algorithm does not rely on nested loops or matrix inversion estimations, which significantly enhances computational efficiency. Additionally, when handling lower-level constraints, our method avoids the need for complex projection steps.

However, we recognize that large-scale applications of bilevel optimization, particularly those with lower-level constraints, remain in their early stages due to the lack of efficient solution tools. We believe that our work provides a step toward developing more scalable and efficient approaches for these complex problems.

**Scope of Current Experiments**

Our experimental evaluation is primarily focused on toy numerical problems and small-scale real-world scenarios. This limitation was partly due to hardware constraints and the challenge of identifying suitable large-scale models that align with our problem framework.

However, we recognize the importance of demonstrating the broader applicability and scalability of BiC-GAFFA. We are actively working on expanding our research to include more comprehensive evaluations on larger and more diverse datasets. This will allow us to better showcase the practical potential and robustness of our algorithm in real-world, large-scale applications.

### A.3 PROOFS IN SECTION 2

### A.3.1 PROOF OF LEMMA 2.1

Lemma 2.1 can be proved using proof techniques similar to those in Theorem 3.3 from Von Heusinger & Kanzow (2009). For completeness, we present an alternative proof of Lemma 2.1 here.

**Lemma 2.1.** *Assume that both $f(x, \cdot)$ and $g(x, \cdot)$ are convex for any given $x \in X$. Let $\gamma_1, \gamma_2 > 0$, we have $\mathcal{G}_\gamma(x, y, z) \geq 0$ for any $(x, y, z) \in X \times Y \times \mathbb{R}^p_+$. Furthermore,*

$$\mathcal{G}_\gamma(x, y, z) \leq 0,$$

*if and only if $y \in S(x)$ and $z \in \mathcal{M}(x, y)$, where $\mathcal{M}(x, y)$ denotes the set of multipliers of the lower-level problem at $(x, y)$, i.e.,*

$$\mathcal{M}(x, y) := \left\{ \lambda \in \mathbb{R}^p_+ \mid 0 \in \nabla_y f(x, y) + \lambda^{\mathrm{T}} \nabla_y g(x, y) + \mathcal{N}_Y(y), \ \lambda^{\mathrm{T}} g(x, y) = 0 \right\}.$$

*Proof.* For any $(x, y, z) \in X \times Y \times \mathbb{R}^p_+$, we have

$$\min_{\theta \in Y} \left\{ f(x, \theta) + z^{\mathrm{T}} g(x, \theta) + \frac{1}{2\gamma_1} \|\theta - y\|^2 \right\}$$
$$\leq f(x, y) + z^{\mathrm{T}} g(x, y) \tag{24}$$
$$\leq \max_{\lambda \in \mathbb{R}^p_+} \left\{ f(x, y) + \lambda^{\mathrm{T}} g(x, y) - \frac{1}{2\gamma_2} \|\lambda - z\|^2 \right\},$$

which implies that

$$\mathcal{G}_\gamma(x, y, z) = \max_{\lambda \in \mathbb{R}^p_+} \left\{ f(x, y) + \lambda^{\mathrm{T}} g(x, y) - \frac{1}{2\gamma_2} \|\lambda - z\|^2 \right\}$$
$$- \min_{\theta \in Y} \left\{ f(x, \theta) + z^{\mathrm{T}} g(x, \theta) + \frac{1}{2\gamma_1} \|\theta - y\|^2 \right\}$$
$$\geq 0.$$

Therefore, $\mathcal{G}_\gamma(x, y, z) = 0$ if and only if

$$\max_{\lambda \in \mathbb{R}^p_+} \left\{ f(x, y) + \lambda^{\mathrm{T}} g(x, y) - \frac{1}{2\gamma_2} \|\lambda - z\|^2 \right\} = \min_{\theta \in Y} \left\{ f(x, \theta) + z^{\mathrm{T}} g(x, \theta) + \frac{1}{2\gamma_1} \|\theta - y\|^2 \right\}.$$

Then, (24) yields that $\mathcal{G}_\gamma(x, y, z) = 0$ if and only if

$$y \in \operatorname*{argmin}_{\theta \in Y} \left\{ f(x, \theta) + z^{\mathrm{T}} g(x, \theta) + \frac{1}{2\gamma_1} \|\theta - y\|^2 \right\},$$
$$z \in \operatorname*{argmax}_{\lambda \in \mathbb{R}^p_+} \left\{ f(x, y) + \lambda^{\mathrm{T}} g(x, y) - \frac{1}{2\gamma_2} \|\lambda - z\|^2 \right\}. \tag{25}$$

Given that the function $f(x, \theta) + z^{\mathrm{T}} g(x, \theta)$ is convex with respect to variable $\theta$, and that $\lambda^{\mathrm{T}} g(x, y)$ is concave with respect to $\lambda$, (25) is equivalent to

$$y \in \operatorname*{argmin}_{\theta \in Y} \left\{ f(x, \theta) + z^{\mathrm{T}} g(x, \theta) \right\},$$
$$z \in \operatorname*{argmax}_{\lambda \in \mathbb{R}^p_+} \left\{ f(x, y) + \lambda^{\mathrm{T}} g(x, y) \right\}, \tag{26}$$

which is equivalent to that $(y, z)$ is a saddle point to

$$\min_{\theta \in Y} \max_{\lambda \in \mathbb{R}^p_+} \ f(x, \theta) + \lambda^{\mathrm{T}} g(x, \theta).$$

Consequently, applying the classical minimax theorem to this convex-concave min-max problem, we obtain that $\mathcal{G}_\gamma(x, y, z) = 0$ if and only if $y \in S(x)$ and $z \in \mathcal{M}(x, y)$. $\qquad\square$

### A.3.2 PROOF OF LEMMA 2.2

**Lemma 2.2.** *Assume that both $f(x, y)$ and $g(x, y)$ are convex in $y$ on $Y$ for any given $x \in X$ and are continuously differentiable on an open set containing $X \times Y$. Then $\mathcal{G}_\gamma(x, y, z)$ is continuously differentiable on $X \times Y \times \mathbb{R}^p_+$, and for any $(x, y, z) \in X \times Y \times \mathbb{R}^p_+$,*

$$\nabla \mathcal{G}_\gamma(x, y, z) = \begin{pmatrix} \nabla_x f(x, y) + (\lambda^*)^{\mathrm{T}} \nabla_x g(x, y) \\ \nabla_y f(x, y) + (\lambda^*)^{\mathrm{T}} \nabla_y g(x, y) \\ -(z - \lambda^*)/\gamma_2 \end{pmatrix} - \begin{pmatrix} \nabla_x f(x, \theta^*) + z^{\mathrm{T}} \nabla_x g(x, \theta^*) \\ (y - \theta^*)/\gamma_1 \\ g(x, \theta^*) \end{pmatrix},$$

*where $\theta^*$ and $\lambda^*$ denote $\theta^*(x, y, z)$ and $\lambda^*(x, y, z)$, respectively, defined as*

$$\theta^*(x, y, z) := \operatorname*{argmin}_{\theta \in Y} \left\{ f(x, \theta) + z^{\mathrm{T}} g(x, \theta) + \frac{1}{2\gamma_1} \|\theta - y\|^2 \right\},$$

$$\lambda^*(x, y, z) := \operatorname*{argmax}_{\lambda \in \mathbb{R}_+^p} \left\{ f(x, y) + \lambda^{\mathrm{T}} g(x, y) - \frac{1}{2\gamma_2} \|\lambda - z\|^2 \right\} = \operatorname{Proj}_{\mathbb{R}_+^p} \left( z + \gamma_2 g(x, y) \right).$$

*Proof.* We first define two auxiliary functions,

$$\mathcal{G}_{1,\gamma}(x, y, z) := \min_{\theta \in Y} \left\{ f(x, \theta) + z^{\mathrm{T}} g(x, \theta) + \frac{1}{2\gamma_1} \|\theta - y\|^2 \right\},$$

$$\mathcal{G}_{2,\gamma}(x, y, z) := \max_{\lambda \in \mathbb{R}_+^p} \left\{ f(x, y) + \lambda^{\mathrm{T}} g(x, y) - \frac{1}{2\gamma_2} \|\lambda - z\|^2 \right\}.$$

Then, it follows that $\mathcal{G}_\gamma(x, y, z) = \mathcal{G}_{2,\gamma}(x, y, z) - \mathcal{G}_{1,\gamma}(x, y, z)$. Since by assumptions that $f$ and $g$ are both continuous differentiable on an open set containing $X \times Y$, it can be easily shown that $f(x, \theta) + z^{\mathrm{T}} g(x, \theta) + \frac{1}{2\gamma_1} \|\theta - y\|^2$ satisfies the inf-compactness condition in Theorem 4.13 of Bonnans & Shapiro (2013) with respect to $\theta \in Y$ on any point $(\bar{x}, \bar{y}, \bar{z}) \in X \times Y \times \mathbb{R}_+^p$, i.e., for any $(\bar{x}, \bar{y}, \bar{z}) \in X \times Y \times \mathbb{R}_+^p$, there exist $c \in \mathbb{R}$, compact set $D$ and neighborhood $W$ of $(\bar{x}, \bar{y}, \bar{z})$ such that the level set $\{\theta \in Y \mid f(x, \theta) + z^{\mathrm{T}} g(x, \theta) + \frac{1}{2\gamma_1} \|\theta - y\|^2 \leq c\}$ is nonempty and contained in $D$ for any $(x, y, z) \in W$. And because of the convexity of $f(x, y)$ and $g(x, y)$ with respect to $y \in Y$ for any $x \in X$, $\operatorname*{argmin}_{\theta \in Y} \left\{ f(x, \theta) + z^{\mathrm{T}} g(x, \theta) + \frac{1}{2\gamma_1} \|\theta - y\|^2 \right\}$ is a singleton for any $(x, y, z) \in X \times Y \times \mathbb{R}_+^p$. Then, by the differentiablility of $f$ and $g$, we can derive from Theorem 4.13, Remark 4.14 of Bonnans & Shapiro (2013) that $\mathcal{G}_{1,\gamma}(x, y, z)$ is differentiable at any point on $X \times Y \times \mathbb{R}_+^p$ and for any $(x, y, z) \in X \times Y \times \mathbb{R}_+^p$,

$$\nabla \mathcal{G}_{1,\gamma}(x, y, z) = \left( \nabla_x f(x, \theta^*) + z^{\mathrm{T}} \nabla_x g(x, \theta^*), (y - \theta^*)/\gamma_1, g(x, \theta^*) \right). \tag{27}$$

By using similar arguments, we can also demonstrate that $\mathcal{G}_{2,\gamma}(x, y, z)$ is differentiable at any point on $X \times Y \times \mathbb{R}_+^p$ and for any $(x, y, z) \in X \times Y \times \mathbb{R}_+^p$,

$$\nabla \mathcal{G}_{2,\gamma}(x, y, z) = \left( \nabla_x f(x, y) + (\lambda^*)^{\mathrm{T}} \nabla_x g(x, y), \nabla_y f(x, y) + (\lambda^*)^{\mathrm{T}} \nabla_y g(x, y), -(z - \lambda^*)/\gamma_2 \right). \tag{28}$$

And then the conclusion follows from $\mathcal{G}_\gamma(x, y, z) = \mathcal{G}_{2,\gamma}(x, y, z) - \mathcal{G}_{1,\gamma}(x, y, z)$. $\qquad\square$

### A.3.3 PROOF OF THEOREM 2.3

**Theorem 2.3.** *Assume that both $f(x, \cdot)$ and $g(x, \cdot)$ are convex for any given $x \in X$. Let $\gamma_1, \gamma_2 > 0$, the reformulation (6) is equivalent to the bilevel optimization problem (1), provided that for any feasible point $(x, y)$ of (1), a corresponding multiplier of the lower-level problem (2) exists at $(x, y)$, i.e., $\mathcal{M}(x, y) \neq \varnothing$.*

*Proof.* Let $(x, y, z)$ be any feasible point of problem (6), then we have $(x, y) \in X \times Y$, $z \in \mathbb{R}_+^p$, $\mathcal{G}_\gamma(x, y, z) \leq 0$. And it follows from Lemma 2.1 that $\mathcal{G}_\gamma(x, y, z) = 0$, $y \in S(x)$ and thus $(x, y)$ is feasible to bilevel program (1).

Now, suppose $(x, y)$ is an feasible point of bilevel program (1), then we have $(x, y) \in X \times Y$ and $y \in S(x)$. According to the assumption that a multiplier $z \in \mathbb{R}_+^p$ of the lower-level problem (2) exists at $(x, y)$, i.e., $z \in \mathcal{M}(x, y)$. Then it follows from Lemma 2.1 that $\mathcal{G}_\gamma(x, y, z) = 0$ and thus $(x, y, z)$ is feasible to reformulation problem (6). $\qquad\square$

## A.4 PROOFS IN SECTION 3

### A.4.1 PROOF OF PROPOSITION 3.1

**Proposition 3.1.** *Suppose $\gamma_1, \gamma_2 > 0$ and an optimal solution $(x^*, y^*, z^*)$ to (6), with $z^* \in Z$, exists, then any optimal solution of (7) is optimal to reformulation (6).*

*Proof.* For any feasible point $(x, y, z)$ of problem (7), $(x, y, z)$ is also feasible to problem (6) and thus the optimal value of problem (7) is larger or equal to that of problem (6). Let $(x^*, y^*, z^*)$ be an optimal solution of reformulation problem (6) with $z^*$ belonging to the set $Z$, then $(x^*, y^*, z^*)$ is also feasible to problem (7). Therefore, the optimal value of problem (7) is equal to that of problem (6). Then, because any feasible point $(x, y, z)$ of problem (7) is feasible to problem (6), we get the conclusion. $\square$

### A.4.2 PROOF OF PROPOSITION 3.2

**Proposition 3.2.** *Assume that $F(x, y)$ is bounded below by $\underline{F}$ on $X \times Y$. For any $\varepsilon > 0$, there exists $\bar{c} > 0$ such that any global solution $(x_c, y_c, z_c)$ to the penalty formulation (8) with penalty parameter $c \geq \bar{c}$ is also a global solution to the relaxed problem (9) with some relaxation parameter $\varepsilon_c$ satisfying $\varepsilon_c \leq \varepsilon$. Moreover, if $(x_c, y_c, z_c)$ is a local solution to the penalty formulation (8), then it is also a local solution to the relaxed problem (9) with relaxation parameter $\varepsilon_c := \mathcal{G}_\gamma(x_c, y_c, z_c)$.*

*Proof.* Let $(\bar{x}, \bar{y}, \bar{z}) \in X \times Y \times Z$ be a feasible point to problem (7) and $(x_c, y_c, z_c) \in X \times Y \times Z$ be a global solution of problem (8) with penalty parameter $c > 0$. We then have

$$F(x_c, y_c) + c\,\mathcal{G}_\gamma(x_c, y_c, z_c) \leq F(\bar{x}, \bar{y}) + c\,\mathcal{G}_\gamma(\bar{x}, \bar{y}, \bar{z}) = F(\bar{x}, \bar{y}),$$

implying

$$\mathcal{G}_\gamma(x_c, y_c, z_c) \leq (F(\bar{x}, \bar{y}) - \underline{F})/c.$$

Thus, for any $\varepsilon > 0$, there exists $\bar{c} > 0$ such that for any $c \geq \bar{c}$,

$$\varepsilon_c := \mathcal{G}_\gamma(x_c, y_c, z_c) \leq \varepsilon.$$

Next, we demonstrate that $(x_c, y_c, z_c)$ is a global solution to problem (9) with relaxation parameter $\varepsilon_c$. Assume, for the sake of contradiction, that there exists $(x, y, z) \in X \times Y \times Z$ with $\mathcal{G}_\gamma(x, y, z) \leq \varepsilon_c$ and $F(x, y) < F(x_c, y_c)$. This leads to

$$F(x, y) + c\,\mathcal{G}_\gamma(x, y, z) < F(x_c, y_c) + c\varepsilon_c = F(x_c, y_c) + c\,\mathcal{G}_\gamma(x_c, y_c, z_c),$$

which contradicts the global optimality of $(x_c, y_c, z_c)$ to problem (8) with penalty parameter $c > 0$.

By analogous reasoning, the assertion concerning the local optimality of $(x_c, y_c, z_c)$ for problem (8) holds similarly. $\square$

### A.4.3 PROOF OF THEOREM 3.3

**Theorem 3.3.** *Assume that $X$ and $Y$ are closed and functions $F$, $f$ and $g$ are continuous on $X \times Y$. Suppose $c_k \to \infty$ and let*

$$(x_k, y_k, z_k) \in \underset{(x,y) \in X \times Y \times Z}{\operatorname{argmin}} \; F(x, y) + c_k\,\mathcal{G}_\gamma(x, y, z).$$

*Then, any accumulation point $(\bar{x}, \bar{y}, \bar{z})$ of the sequence $\{(x_k, y_k, z_k)\}$ is a solution to problem (7).*

*Proof.* Applying the proof techniques used in Lemma 2 of Liu et al. (2020), we can establish that $\mathcal{G}_\gamma(x, y, z)$ is lower semi-continuous on $X \times Y \times Z$. The theorem's conclusion follows by employing the same proof techniques from Theorem 1 of Liu et al. (2020). $\square$

### A.5 PROOFS IN SECTION 4

The proof of non-asymptotic convergence for BiC-GAFFA primarily hinges on establishing the sufficient descent property of the merit function defined as follows

$$V_k := \phi_{c_k}(x^k, y^k, z^k) + C_\theta \left\| \theta^k - \theta^*(x^k, y^k, z^k) \right\|^2,$$

where

$$\phi_{c_k}(x, y, z) := \frac{1}{c_k}\big(F(x, y) - \underline{F}\big) + \mathcal{G}_\gamma(x, y, z),$$

and

$$C_\theta := (L_f + rL_{g_1} + \frac{1}{\gamma_1} + L_g)^2.$$

To establish the sufficiently decreasing property of the merit function, we initially derive several crucial auxiliary lemmas. These lemmas establish the Lipschitz continuity of $\theta^*(x, y, z)$ and $\lambda^*(x, y, z)$ (as detailed in Lemma A.1), the Lipschitz continuity of $\nabla\mathcal{G}_\gamma(x, y, z)$ (as detailed in Lemma A.2) and a descent property with bounded errors for the function $\phi_{c_k}(x, y, z)$ at each iteration (as detailed in Lemma A.4).

### A.5.1 AUXILIARY LEMMAS

The Lipschitz properties of $\theta^*(x, y, z)$, $\lambda^*(x, y, z)$, and $\nabla\mathcal{G}_\gamma(x, y, z)$ are crucial for the convergence analysis. We establish these properties in the subsequent lemmas.

**Lemma A.1.** *Under Assumptions 4.2 and 4.3, and let $\gamma_1 > 0$, $\gamma_2 > 0$, for any $(x, y, z), (x', y', z') \in X \times Y \times Z$, the following inequalities hold*

$$
\begin{aligned}
\|\theta^*(x, y, z) - \theta^*(x', y', z')\| &\leq (\gamma_1 L_f + \gamma_1 r L_{g_2}) \|x - x'\| + \|y - y'\| + \gamma_1 L_g \|z - z'\| \\
&\leq L_\theta \|(x, y, z) - (x', y', z')\|, \\
\|\lambda^*(x, y, z) - \lambda^*(x', y', z')\| &\leq \gamma_2 L_g \|(x, y) - (x', y')\| + \|z - z'\| \\
&\leq L_\lambda \|(x, y, z) - (x', y', z')\|,
\end{aligned}
\tag{29}
$$

*where $L_\theta := \sqrt{3} \max\{\gamma_1 L_f + \gamma_1 r L_{g_2}, 1, \gamma_1 L_g\}$ and $L_\lambda := \sqrt{2} \max\{\gamma_2 L_g, 1\}$.*

*Proof.* To simplify notation, we denote $(x, y, z), (x', y', z') \in X \times Y \times Z$ by $w$ and $w'$, respectively. Considering that $\theta^*(w)$ and $\lambda^*(w)$ are optimal solutions to optimization problems in (5), it follows from the stationary conditions that

$$
\begin{aligned}
0 &\in \nabla_y f(x, \theta^*(w)) + z^{\mathrm{T}} \nabla_y g(x, \theta^*(w)) + (\theta^*(w) - y)/\gamma_1 + \mathcal{N}_Y(\theta^*(w)), \\
0 &\in -g(x, y) + (\lambda^*(w) - z)/\gamma_2 + \mathcal{N}_{\mathbb{R}_+^p}(\lambda^*(w)).
\end{aligned}
\tag{30}
$$

Same results apply to $\theta^*(w')$ and $\lambda^*(w')$

$$
\begin{aligned}
0 &\in \nabla_y f(x', \theta^*(w')) + (z')^{\mathrm{T}} \nabla_y g(x', \theta^*(w')) + (\theta^*(w') - y')/\gamma_1 + \mathcal{N}_Y(\theta^*(w')), \\
0 &\in -g(x', y') + (\lambda^*(w') - z')/\gamma_2 + \mathcal{N}_{\mathbb{R}_+^p}(\lambda^*(w')).
\end{aligned}
\tag{31}
$$

Defining

$$
T(x, y, z, \theta) := \nabla_\theta \left( f(x, \theta) + z^{\mathrm{T}} g(x, \theta) + \frac{1}{2\gamma_1} \|\theta - y\|^2 \right).
$$

and exploiting the monotonicity of $\mathcal{N}_Y$, we have from (30) and (31) that

$$
\begin{aligned}
\langle - T(w, \theta^*(w)) + T(w, \theta^*(w')), \theta^*(w) - \theta^*(w') \rangle \\
+ \langle - T(w, \theta^*(w')) + T(w', \theta^*(w')), \theta^*(w) - \theta^*(w') \rangle \geq 0
\end{aligned}
\tag{32}
$$

Under Assumptions 4.2 and 4.3, and given that $\gamma_1 > 0$, it holds that for any $(x, y, z) \in X \times Y \times \mathbb{R}_+^p$,

$$
f(x, \theta) + z^{\mathrm{T}} g(x, \theta) + \frac{1}{2\gamma_1} \|\theta - y\|^2
$$

is $\frac{1}{\gamma_1}$-strongly convex with respect to $\theta$. According to Rockafellar & Wets (2009), $T(x, y, z, \theta)$ is $1/\gamma_1$-strongly monotone. Consequently, we have that

$$
\langle T(w, \theta^*(w)) - T(w, \theta^*(w')), \theta^*(w) - \theta^*(w') \rangle \geq \|\theta^*(w) - \theta^*(w')\|^2/\gamma_1.
\tag{33}
$$

Under Assumptions 4.2 and 4.3, we establish that

$$
\begin{aligned}
&\| - T(w, \theta^*(w')) + T(w', \theta^*(w'))\| \\
&\leq \|\nabla_y f(x, \theta^*(w')) - \nabla_y f(x', \theta^*(w'))\| + \|z^{\mathrm{T}} \nabla_y g(x, \theta^*(w')) - (z')^{\mathrm{T}} \nabla_y g(x', \theta^*(w'))\| + \|y - y'\|/\gamma_1 \\
&\leq L_f \|x - x'\| + \|z^{\mathrm{T}} \nabla_y g(x, \theta^*(w')) - z^{\mathrm{T}} \nabla_y g(x', \theta^*(w'))\| \\
&\quad + \|z^{\mathrm{T}} \nabla_y g(x', \theta^*(w')) - (z')^{\mathrm{T}} \nabla_y g(x', \theta^*(w'))\| + \|y - y'\|/\gamma_1 \\
&\leq L_f \|x - x'\| + r L_{g_2} \|x - x'\| + L_g \|z - z'\| + \|y - y'\|/\gamma_1,
\end{aligned}
$$

where the last inequality follows from the $L_{g_2}$-Lipschitz continuity of $\nabla_y g$ on $X \times Y$, $\max_{x \in X, y \in Y} \|\nabla_y g(x,y)\| \leq L_g$, $(x', \theta^*(w')) \in X \times Y$ and $z \in Z$. Combining this inequality with (32) and (33), we have

$$\|\theta^*(w) - \theta^*(w')\| \leq (\gamma_1 L_f + \gamma_1 r L_{g_2}) \|x - x'\| + \|y - y'\| + \gamma_1 L_g \|z - z'\|.$$

Further, exploiting the monotonicity of $\mathcal{N}_{\mathbb{R}_+^p}$, we obtain from (30) and (31) that

$$\langle g(x,y) + (z - \lambda^*(w))/\gamma_2 - g(x',y') - (z' - \lambda^*(w'))/\gamma_2, \lambda^*(w) - \lambda^*(w') \rangle \geq 0. \qquad (34)$$

Then, invoking Assumption 4.3, we have that

$$\|\lambda^*(w) - \lambda^*(w')\| \leq \gamma_2 L_g \|(x,y) - (x',y')\| + \|z - z'\|.$$

$\square$

**Lemma A.2.** *Under Assumptions 4.2 and 4.3, assume that $X$, $Y$ are compact sets, and $\gamma_1 > 0$, $\gamma_2 > 0$. Then, there exists $L_{\mathcal{G}} > 0$ such that for any points $(x,y,z), (x',y',z') \in X \times Y \times Z$,*

$$\|\nabla \mathcal{G}_\gamma(x,y,z) - \nabla \mathcal{G}_\gamma(x',y',z')\| \leq L_{\mathcal{G}} \|(x,y,z) - (x',y',z')\|,$$

*and*

$$\mathcal{G}_\gamma(x',y',z') \leq \mathcal{G}_\gamma(x,y,z) + \langle \nabla \mathcal{G}_\gamma(x,y,z), (x',y',z') - (x,y,z) \rangle + \frac{L_{\mathcal{G}}}{2} \|(x,y,z) - (x',y',z')\|^2.$$

*Proof.* For conciseness, we denote $(x,y,z), (x',y',z') \in X \times Y \times Z$ by $w$ and $w'$, respectively. Recalling from Lemma A.1, we have

$$\|\theta^*(w) - \theta^*(w')\| \leq L_\theta \|w - w'\|, \qquad \|\lambda^*(w) - \lambda^*(w')\| \leq L_\lambda \|w - w'\|. \qquad (35)$$

As specified in (5), the norm of $\lambda^*(w)$ is bounded above by

$$\|\lambda^*(w)\| = \|\mathrm{Proj}_{\mathbb{R}_+^p}(z + \gamma_2 g(x,y))\| \leq \|z + \gamma_2 g(x,y)\| \leq r + \gamma_2 M_g, \qquad \forall w \in X \times Y \times Z,$$

where $M_g := \max_{x \in X, y \in Y} \|g(x,y)\|$. Drawing upon Lemma 2.2, Assumptions 4.2 and 4.3, and (35), for any $w, w' \in X \times Y \times Z$, we have

$$\|\nabla_x \mathcal{G}_\gamma(w) - \nabla_x \mathcal{G}_\gamma(w')\|$$
$$\leq \|\nabla_x f(x,y) - \nabla_x f(x',y')\| + \|\lambda^*(w)^{\mathrm{T}} \nabla_x g(x,y) - \lambda^*(w')^{\mathrm{T}} \nabla_x g(x',y')\|$$
$$\quad + \|\nabla_x f(x, \theta^*(w)) - \nabla_x f(x', \theta^*(w'))\| + \|z^{\mathrm{T}} \nabla_x g(x, \theta^*(w)) - (z')^{\mathrm{T}} \nabla_x g(x', \theta^*(w'))\|$$
$$\leq L_f \|(x,y) - (x',y')\| + L_f \|(x, \theta^*(w)) - (x', \theta^*(w'))\|$$
$$\quad + \|\lambda^*(w)^{\mathrm{T}} \nabla_x g(x,y) - \lambda^*(w)^{\mathrm{T}} \nabla_x g(x',y')\| + \|\lambda^*(w)^{\mathrm{T}} \nabla_x g(x',y') - \lambda^*(w')^{\mathrm{T}} \nabla_x g(x',y')\|$$
$$\quad + \|z^{\mathrm{T}} \nabla_x g(x, \theta^*(w)) - z^{\mathrm{T}} \nabla_x g(x', \theta^*(w'))\| + \|z^{\mathrm{T}} \nabla_x g(x', \theta^*(w')) - (z')^{\mathrm{T}} \nabla_x g(x', \theta^*(w'))\|$$
$$\leq L_f \|(x,y) - (x',y')\| + L_f \|x - x'\| + L_f L_\theta \|w - w'\|$$
$$\quad + (r + \gamma_2 M_g) L_{g_1} \|(x,y) - (x',y')\| + L_g L_\lambda \|w - w'\|$$
$$\quad + r L_{g_1} \|x - x'\| + r L_{g_1} L_\theta \|w - w'\| + L_g \|z - z'\|,$$

where the last inequality follows from $\theta^*(w') \in Y$, $z \in Z$ and $\max_{x \in X, y \in Y} \|\nabla_x g(x,y)\| \leq L_g$. Similarly, for gradients with respect to $y$ and $z$, for any $w, w' \in X \times Y \times Z$, we have

$$\|\nabla_y \mathcal{G}_\gamma(w) - \nabla_y \mathcal{G}_\gamma(w')\|$$
$$\leq \|\nabla_y f(x,y) - \nabla_y f(x',y')\| + \|\lambda^*(w)^{\mathrm{T}} \nabla_y g(x,y) - \lambda^*(w)^{\mathrm{T}} \nabla_y g(x',y')\|$$
$$\quad + \|\lambda^*(w)^{\mathrm{T}} \nabla_y g(x',y') - \lambda^*(w')^{\mathrm{T}} \nabla_y g(x',y')\| + \frac{1}{\gamma_1} \|y - y'\| + \frac{1}{\gamma_1} \|\theta^*(w) - \theta^*(w')\|$$
$$\leq L_f \|(x,y) - (x',y')\| + (r + \gamma_2 M_g) L_{g_2} \|(x,y) - (x',y')\|$$
$$\quad + L_g L_\lambda \|w - w'\| + \frac{1}{\gamma_1} \|y - y'\| + \frac{1}{\gamma_1} L_\theta \|w - w'\|,$$

where the last inequality follows from the fact that $\max_{x \in X, y \in Y} \|\nabla_y g(x, y)\| \leq L_g$, and

$$
\|\nabla_z \mathcal{G}_\gamma(w) - \nabla_z \mathcal{G}_\gamma(w')\|
$$
$$
\leq \frac{1}{\gamma_2}\|z - z'\| + \frac{1}{\gamma_2}\|\lambda^*(w) - \lambda^*(w')\| + \|g(x, \theta^*(w)) - g(x', \theta^*(w'))\|
$$
$$
\leq \frac{1}{\gamma_2}\|z - z'\| + \frac{L_\lambda}{\gamma_2}\|w - w'\| + L_g\|x - x'\| + L_g L_\theta \|w - w'\|.
$$

The above inequalities together yields the existence of constant $L_{\mathcal{G}} > 0$ such that

$$
\|\nabla \mathcal{G}_\gamma(x, y, z) - \nabla \mathcal{G}_\gamma(x', y', z')\| \leq L_{\mathcal{G}}\|(x, y, z) - (x', y', z')\|.
$$

Then the conclusion follows from Lemma 5.7 of Beck (2017). $\qquad \square$

The update of $\theta^k$ constitutes a single gradient descent step to the minimization problem defined in (5). The progress of this update is quantified in the subsequent lemma.

**Lemma A.3.** *Under Assumptions 4.2 and 4.3, let $\gamma_1 > 0$, $\gamma_2 > 0$ and $\eta_k \in (0, 1/(L_f + rL_{g_2} + 1/\gamma_1))$, then the sequence of $(x^k, y^k, z^k, \theta^k, \lambda^k)$ generated by Algorithm 1 satisfies*

$$
\|\theta^{k+1} - \theta^*(x^k, y^k, z^k)\|^2 \leq (1 - \eta_k/\gamma_1)\|\theta^k - \theta^*(x^k, y^k, z^k)\|^2, \qquad (36)
$$

*and*

$$
\lambda^{k+1} = \lambda^*(x^k, y^k, z^k).
$$

*Proof.* Consider $(x^k, y^k, z^k) \in X \times Y \times Z$. For brevity, we denote $\theta^*(x^k, y^k, z^k)$ and $\lambda^*(x^k, y^k, z^k)$ by $\theta^*$ and $\lambda^*$, respectively. Under Assumptions 4.2 and 4.3, and given $\gamma_1 > 0$, the function

$$
f(x^k, \theta) + (z^k)^\mathrm{T} g(x^k, \theta) + \frac{1}{2\gamma_1}\|\theta - y^k\|^2,
$$

is $\frac{1}{\gamma_1}$-strongly convex and $(L_f + rL_{g_2} + 1/\gamma_1)$-smooth with respect to $\theta$. invoking Theorem 10.29 of Beck (2017), we obtain

$$
\|\theta^{k+1} - \theta^*\|^2 \leq (1 - \eta_k/\gamma_1)\|\theta^k - \theta^*\|^2.
$$

Additionally, according to (5), it follows that $\lambda^{k+1} = \lambda^*$. $\qquad \square$

The update of variables $(x, y, z)$ in (14) can be viewed as an inexact alternating proximal gradient step from $(x^k, y^k, z^k)$ on solving $\min_{(x,y,z) \in X \times Y \times Z} \phi_{c_k}(x, y, z)$. In the lemma below, we demonstrate that the function $\phi_{c_k}(x, y, z)$ exhibits a decreasing property with bounded errors at each iteration.

**Lemma A.4.** *Under Assumptions 4.2 and 4.3, assume $X$, $Y$ are compact sets, and let $\gamma_1 > 0$, $\gamma_2 > 0$. Then the sequence of $(x^k, y^k, z^k, \theta^k, \lambda^k)$ generated by Algorithm 1 satisfies*

$$
\phi_{c_k}(x^{k+1}, y^{k+1}, z^{k+1}) \leq \phi_{c_k}(x^k, y^k, z^k) - \left(\frac{1}{2\alpha_k} - \frac{L_{\phi_k}}{2}\right)\|(x^{k+1}, y^{k+1}, z^{k+1}) - (x^k, y^k, z^k)\|^2
$$
$$
+ \frac{\alpha_k}{2}\left(L_f + u_z L_{g_1} + \frac{1}{\gamma_1} + L_g\right)\left\|\theta^*(x^k, y^k, z^k) - \theta^{k+1}\right\|^2.
$$

$$(37)$$

*where $L_{\phi_k} := L_F/c_k + L_{\mathcal{G}}$.*

*Proof.* For clarity, we denote $(x^k, y^k, z^k)$, $(x^{k+1}, y^{k+1}, z^{k+1})$ as $w^k$ and $w^{k+1}$, respectively. Under the Assumptions 4.2 and 4.3, where $\nabla F$ and $\nabla f$ are $L_F$- and $L_f$-Lipschitz continuous on $X \times Y$, and applying Lemma 5.7 of Beck (2017) and Lemma A.2, we obtain

$$
\phi_{c_k}(x^{k+1}, y^{k+1}, z^{k+1}) \leq \phi_{c_k}(x^k, y^k, z^k) + \langle \nabla \phi_{c_k}(x^k, y^k, z^k), (x^{k+1}, y^{k+1}, z^{k+1}) - (x^k, y^k, z^k)\rangle
$$
$$
+ \frac{L_{\phi_k}}{2}\|(x^{k+1}, y^{k+1}, z^{k+1}) - (x^k, y^k, z^k)\|^2,
$$

$$(38)$$

with $L_{\phi_k} := L_F/c_k + L_{\mathcal{G}}$. Based on the update rule of variables $(x, y, z)$ in (14), the convexity of $X \times Y \times Z$ and the property of the projection operator $\mathrm{Proj}_{X \times Y \times Z}$, we have

$$\left\langle (x^k, y^k, z^k) - \alpha_k(d_x^k, d_y^k, d_z^k) - (x^{k+1}, y^{k+1}, z^{k+1}), (x^k, y^k, z^k) - (x^{k+1}, y^{k+1}, z^{k+1}) \right\rangle \leq 0,$$

yielding

$$\left\langle (d_x^k, d_y^k, d_z^k), (x^{k+1}, y^{k+1}, z^{k+1}) - (x^k, y^k, z^k) \right\rangle \leq -\frac{1}{\alpha_k} \|(x^{k+1}, y^{k+1}, z^{k+1}) - (x^k, y^k, z^k)\|^2.$$

Integrating this inequality with (38), we infer that

$$\phi_{c_k}(x^{k+1}, y^{k+1}, z^{k+1}) \leq \phi_{c_k}(x^k, y^k, z^k) - \left( \frac{1}{\alpha_k} - \frac{L_{\phi_k}}{2} \right) \|(x^{k+1}, y^{k+1}, z^{k+1}) - (x^k, y^k, z^k)\|^2$$
$$+ \left\langle \nabla \phi_{c_k}(x^k, y^k, z^k) - (d_x^k, d_y^k, d_z^k), (x^{k+1}, y^{k+1}, z^{k+1}) - (x^k, y^k, z^k) \right\rangle.$$
$$(39)$$

Furthermore, considering $w^k \in X \times Y \times Z$ and with $\theta^*(w^k), \theta^k \in Y$ for all $k$, and utilizing the formula of $\nabla \mathcal{G}_\gamma(x, y, z)$ derived in Lemma 2.2, along with the definitions of $d_x^k$, $d_y^k$ and $d_z^k$ in (13) and $\lambda^{k+1} = \lambda^*(w^k)$ from Lemma A.3, we can obtain from Assumptions 4.2 and 4.3 that

$$\left\| \nabla \phi_{c_k}(x^k, y^k, z^k) - (d_x^k, d_y^k, d_z^k) \right\|$$
$$\leq \left\| \nabla_x \mathcal{G}_\gamma(x^k, y^k, z^k) - \nabla_x f(x^k, y^k) - (\lambda^{k+1})^{\mathrm{T}} \nabla_x g(x^k, y^k) + \nabla_x f(x^k, \theta^{k+1}) + (z^k)^{\mathrm{T}} \nabla_x g(x^k, \theta^{k+1}) \right\|$$
$$+ \left\| \nabla_y \mathcal{G}_\gamma(x^k, y^k, z^k) - \nabla_y f(x^k, y^k) - (\lambda^{k+1})^{\mathrm{T}} \nabla_y g(x^k, y^k) + (y^k - \theta^{k+1})/\gamma_1 \right\|$$
$$+ \left\| \nabla_z \mathcal{G}_\gamma(x^k, y^k, z^k) + (z^k - \lambda^{k+1})/\gamma_2 + g(x^k, \theta^{k+1}) \right\|$$
$$\leq \left\| \lambda^*(w^k)^{\mathrm{T}} \nabla_x g(x^k, y^k) - (\lambda^{k+1})^{\mathrm{T}} \nabla_x g(x^k, y^k) \right\| + \left\| \nabla_x f(x^k, \theta^*(w^k)) - \nabla_x f(x^k, \theta^{k+1}) \right\|$$
$$+ \left\| (z^k)^{\mathrm{T}} \nabla_x g(x^k, \theta^*(w^k)) - (z^k)^{\mathrm{T}} \nabla_x g(x^k, \theta^{k+1}) \right\|$$
$$+ \left\| \lambda^*(w^k)^{\mathrm{T}} \nabla_y g(x^k, y^k) - (\lambda^{k+1})^{\mathrm{T}} \nabla_y g(x^k, y^k) \right\| + \frac{1}{\gamma_1} \left\| \theta^*(w^k) - \theta^{k+1} \right\|$$
$$+ \frac{1}{\gamma_2} \left\| \lambda^*(w^k) - \lambda^{k+1} \right\| + \left\| g(x^k, \theta^*(w^k)) - g(x^k, \theta^{k+1}) \right\|$$
$$\leq L_f \left\| \theta^*(w^k) - \theta^{k+1} \right\| + rL_{g_1} \left\| \theta^*(w^k) - \theta^{k+1} \right\| + \frac{1}{\gamma_1} \left\| \theta^*(w^k) - \theta^{k+1} \right\| + L_g \left\| \theta^*(w^k) - \theta^{k+1} \right\|$$
$$= \left( L_f + rL_{g_1} + \frac{1}{\gamma_1} + L_g \right) \left\| \theta^*(w^k) - \theta^{k+1} \right\|.$$
$$(40)$$

This yields that

$$\left\langle \nabla \phi_{c_k}(x^k, y^k, z^k) - (d_x^k, d_y^k, d_z^k), (x^{k+1}, y^{k+1}, z^{k+1}) - (x^k, y^k, z^k) \right\rangle$$
$$\leq \frac{\alpha_k}{2} \left( L_f + rL_{g_1} + \frac{1}{\gamma_1} + L_g \right)^2 \left\| \theta^*(w^k) - \theta^{k+1} \right\|^2 + \frac{1}{2\alpha_k} \|(x^{k+1}, y^{k+1}, z^{k+1}) - (x^k, y^k, z^k)\|^2,$$

Combining this with (39) leads to the following inequality

$$\phi_{c_k}(x^{k+1}, y^{k+1}, z^{k+1}) \leq \phi_{c_k}(x^k, y^k, z^k) - \left( \frac{1}{2\alpha_k} - \frac{L_{\phi_k}}{2} \right) \|(x^{k+1}, y^{k+1}, z^{k+1}) - (x^k, y^k, z^k)\|^2$$
$$+ \frac{\alpha_k}{2} \left( L_f + rL_{g_1} + \frac{1}{\gamma_1} + L_g \right)^2 \left\| \theta^*(w^k) - \theta^{k+1} \right\|^2.$$
$$(41)$$
$$\square$$

### A.5.2 SUFFICIENT DESCENT PROPERTY OF $V_k$

Utilizing the auxiliary lemmas established previously, we now proceed to demonstrate the sufficient decreasing property of $V_k$.

**Lemma A.5.** *Under Assumptions 4.1, 4.2 and 4.3, suppose $X$, $Y$ are compact sets, $\gamma_1 > 0$, $\gamma_2 > 0$ and $\eta_k \in (\underline{\eta}, 1/(L_f + rL_{g_2} + 1/\gamma_1))$ with $\underline{\eta} > 0$. Then there exists $c_\alpha > 0$ such that when $0 < \alpha_k \le c_\alpha$, the sequence of $(x^k, y^k, z^k, \theta^k, \lambda^k)$ generated by Algorithm 1 satisfies*

$$V_{k+1} - V_k \le -\frac{1}{4\alpha_k} \left\| w^{k+1} - w^k \right\|^2 - \frac{\eta_k C_\theta}{2\gamma_1} \|\theta^k - \theta^*(w^k)\|^2. \tag{42}$$

*Proof.* For clarity and conciseness, we represent $(x^k, y^k, z^k), (x^{k+1}, y^{k+1}, z^{k+1})$ as $w^k$ and $w^{k+1}$, respectively. Recall (37) from Lemma A.4 that

$$\phi_{c_k}(x^{k+1}, y^{k+1}, z^{k+1}) \le \phi_{c_k}(x^k, y^k, z^k) - \left( \frac{1}{2\alpha_k} - \frac{L_{\phi_k}}{2} \right) \|(x^{k+1}, y^{k+1}, z^{k+1}) - (x^k, y^k, z^k)\|^2$$

$$+ \frac{\alpha_k}{2} \left( L_f + rL_{g_1} + \frac{1}{\gamma_1} + L_g \right)^2 \left\| \theta^*(x^k, y^k, z^k) - \theta^{k+1} \right\|^2. \tag{43}$$

Given that $c_{k+1} \ge c_k$, it follows that $(F(x^{k+1}, y^{k+1}) - \underline{F})/c_{k+1} \le (F(x^{k+1}, y^{k+1}) - \underline{F})/c_k$. Combining this with (43) leads to

$$V_{k+1} - V_k = \phi_{c_{k+1}}(w^{k+1}) - \phi_{c_k}(w^k) + C_\theta \left\| \theta^{k+1} - \theta^*(w^{k+1}) \right\|^2 - C_\theta \left\| \theta^k - \theta^*(w^k) \right\|^2$$

$$\le \phi_{c_k}(w^{k+1}) - \phi_{c_k}(w^k) + C_\theta \left\| \theta^{k+1} - \theta^*(w^{k+1}) \right\|^2 - C_\theta \left\| \theta^k - \theta^*(w^k) \right\|^2$$

$$\le - \left( \frac{1}{2\alpha_k} - \frac{L_{\phi_k}}{2} \right) \|w^{k+1} - w^k\|^2 + C_\theta \left\| \theta^{k+1} - \theta^*(w^{k+1}) \right\|^2 - C_\theta \left\| \theta^k - \theta^*(w^k) \right\|^2$$

$$+ \frac{\alpha_k}{2} \left( L_f + rL_{g_1} + \frac{1}{\gamma_1} + L_g \right)^2 \left\| \theta^*(w^k) - \theta^{k+1} \right\|^2$$

$$= - \left( \frac{1}{2\alpha_k} - \frac{L_{\phi_k}}{2} \right) \|w^{k+1} - w^k\|^2 + C_\theta \left\| \theta^{k+1} - \theta^*(w^{k+1}) \right\|^2 - C_\theta \left\| \theta^k - \theta^*(w^k) \right\|^2$$

$$+ \frac{\alpha_k}{2} C_\theta \left\| \theta^*(w^k) - \theta^{k+1} \right\|^2 \tag{44}$$

where the last equation follows from defining $C_\theta := (L_f + rL_{g_1} + \frac{1}{\gamma_1} + L_g)^2$. Further, we can demonstrate that

$$\left\| \theta^{k+1} - \theta^*(w^{k+1}) \right\|^2 - \left\| \theta^k - \theta^*(w^k) \right\|^2 + \frac{\alpha_k}{2} \left\| \theta^*(w^k) - \theta^{k+1} \right\|^2$$

$$\le (1 + \epsilon_k + \alpha_k/2) \left\| \theta^{k+1} - \theta^*(w^k) \right\|^2 - \left\| \theta^k - \theta^*(w^k) \right\|^2 + (1 + \frac{1}{\epsilon_k}) \|\theta^*(w^{k+1}) - \theta^*(w^k)\|^2$$

$$\le (1 + \epsilon_k + \alpha_k/2)(1 - \eta_k/\gamma_1) \|\theta^k - \theta^*(w^k)\|^2 - \left\| \theta^k - \theta^*(w^k) \right\|^2 + (1 + \frac{1}{\epsilon_k}) L_\theta^2 \left\| w^{k+1} - w^k \right\|^2,$$

for any $\epsilon_k > 0$, where the second inequality is a consequence of Lemmas A.1 and A.3. By setting $\epsilon_k = \frac{1}{4}\eta_k/\gamma_1$ in the above inequality, we obtain that when $\alpha_k \le \frac{1}{2}\eta_k/\gamma_1$, the following inequalities hold

$$(1 + \epsilon_k + \alpha_k/2)(1 - \eta_k/\gamma_1) \le (1 + \frac{1}{2}\eta_k/\gamma_1)(1 - \eta_k/\gamma_1) \le 1 - \frac{\eta_k}{2\gamma_1}.$$

Consequently, we establish the inequality

$$\left\| \theta^{k+1} - \theta^*(w^{k+1}) \right\|^2 - \left\| \theta^k - \theta^*(w^k) \right\|^2 + \frac{\alpha_k}{2} \left\| \theta^*(w^k) - \theta^{k+1} \right\|^2$$

$$\le -\frac{\eta_k}{2\gamma_1} \|\theta^k - \theta^*(w^k)\|^2 + \left( 1 + \frac{4\gamma_1}{\eta_k} \right) L_\theta^2 \left\| w^{k+1} - w^k \right\|^2. \tag{45}$$

Combining (44) and (45) implies

$$V_{k+1} - V_k \le - \left[ \frac{1}{2\alpha_k} - \frac{L_{\phi_k}}{2} - \left( 1 + \frac{4\gamma_1}{\eta_k} \right) L_\theta^2 C_\theta \right] \left\| w^{k+1} - w^k \right\|^2 - \frac{\eta_k C_\theta}{2\gamma_1} \|\theta^k - \theta^*(w^k)\|^2. \tag{46}$$

When $c_{k+1} \geq c_k$, $\eta_k \geq \underline{\eta} > 0$, $\alpha_k \leq \frac{1}{2}\underline{\eta}/\gamma_1$, it holds that for any $k$, $\alpha_k \leq \frac{1}{2}\eta_k/\gamma_1$,

$$\frac{L_{\phi_k}}{2} + \left(1 + \frac{4\gamma_1}{\eta_k}\right) L_\theta^2 C_\theta \leq \frac{L_{\phi_0}}{2} + \left(1 + \frac{4\gamma_1}{\underline{\eta}}\right) L_\theta^2 C_\theta =: C_\alpha.$$

If $c_\alpha > 0$ satisfies

$$c_\alpha \leq \min\left\{\frac{1}{2}\underline{\eta}/\gamma_1, \frac{1}{4C_\alpha}\right\},$$

then, for $0 < \alpha_k \leq c_\alpha$, it is guaranteed that

$$\frac{1}{2\alpha_k} - \frac{L_{\phi_k}}{2} - \left(1 + \frac{4\gamma_1}{\eta_k}\right) L_\theta^2 C_\theta \geq \frac{1}{4\alpha_k}.$$

Therefore, the conclusion follows directly from (46). □

### A.5.3 PROOFS OF THEOREM 4.4 AND THEOREM 4.5

Indeed, Theorem 4.4 is a special case of Theorem 4.5 with $\rho = 0$. Consequently, we present the proof of Theorem 4.5.

**Theorem 4.5.** *Under Assumptions 4.1, 4.2 and 4.3, assume that $X$, $Y$ are compact sets, $\gamma_1 > 0$, $\gamma_2 > 0$, $c_k = c(k+1)^\rho$ with $c > 0$, $\rho \in [0, 1/2)$ and $\eta_k \in (\underline{\eta}, 1/(L_f + rL_{g_2} + 1/\gamma_1))$ with $\underline{\eta} > 0$. Then there exists $c_\alpha > 0$ such that when $\alpha_k \in (\underline{\alpha}, c_\alpha)$ with $\underline{\alpha} > 0$, the sequence of $(x^k, y^k, z^k, \theta^k, \lambda^k)$ generated by Algorithm 1 satisfies*

$$\min_{0 \leq k \leq K} \left\|\theta^k - \theta^*(x^k, y^k, z^k)\right\| = O\left(\frac{1}{K^{1/2}}\right),$$

*and*

$$\min_{0 \leq k \leq K} R_k(x^{k+1}, y^{k+1}, z^{k+1}) = O\left(\frac{1}{K^{(1-2\rho)/2}}\right).$$

*Furthermore, if $\rho > 0$ and $\psi_{c_k}(x^k, y^k, z^k)$ is uniformly bounded above, then the sequence of $(x^k, y^k, z^k)$ satisfies*

$$0 \leq \mathcal{G}_\gamma(x^K, y^K, z^K) = O\left(\frac{1}{K^\rho}\right).$$

*Proof.* Lemma A.5 establishes the existence of $c_\alpha > 0$ such that (42) holds when $\alpha_k \leq c_\alpha$. Summing (42) over $k = 0, 1, \ldots, K-1$, we obtain

$$\sum_{k=0}^{K-1} \left(\frac{1}{4\alpha_k}\|(x^{k+1}, y^{k+1}, z^{k+1}) - (x^k, y^k, z^k)\|^2 + \frac{\eta C_\theta}{2\gamma_1}\|\theta^k - \theta^*(x^k, y^k, z^k)\|^2\right) \tag{47}$$
$$\leq V_0 - V_K \leq V_0,$$

where the last inequality follows from the nonnegativity of $V_K$. Consequently, it holds that

$$\sum_{k=0}^{\infty} \|\theta^k - \theta^*(x^k, y^k, z^k)\|^2 < \infty,$$

and

$$\min_{0 \leq k \leq K} \left\|\theta^k - \theta^*(x^k, y^k, z^k)\right\| = O\left(\frac{1}{K^{1/2}}\right).$$

According to the update rule for variables $(x, y, z)$ in (14), we derive

$$0 \in c_k(d_x^k, d_y^k, d_z^k) + \mathcal{N}_{X \times Y \times Z}(x^{k+1}, y^{k+1}, z^{k+1}) + \frac{c_k}{\alpha_k}\left((x^{k+1}, y^{k+1}, z^{k+1}) - (x^k, y^k, z^k)\right). \tag{48}$$

Following the definitions of $d_x^k$, $d_y^k$ and $d_z^k$ in (13), we obtain

$$e^k \in \nabla\psi_{c_k}(x^{k+1}, y^{k+1}, z^{k+1}) + \mathcal{N}_{X \times Y \times Z}(x^{k+1}, y^{k+1}, z^{k+1}),$$

where

$$e^k := \nabla\psi_{c_k}(x^{k+1}, y^{k+1}, z^{k+1}) - c_k(d_x^k, d_y^k, d_z^k) - \frac{c_k}{\alpha_k}\left((x^{k+1}, y^{k+1}, z^{k+1}) - (x^k, y^k, z^k)\right). \quad (49)$$

Next, we estimate $\|e^k\|$. We have

$$\|e^k\| \le \|\nabla\psi_{c_k}(x^{k+1}, y^{k+1}, z^{k+1}) - \nabla\psi_{c_k}(x^k, y^k, z^k)\| + \|\nabla\psi_{c_k}(x^k, y^k, z^k) - c_k(d_x^k, d_y^k, d_z^k)\|$$
$$+ \frac{c_k}{\alpha_k}\left\|(x^{k+1}, y^{k+1}, z^{k+1}) - (x^k, y^k, z^k)\right\|.$$

For the first term in the right hand side of the above inequality, by using Assumptions 4.1, 4.2 and 4.3, along with Lemma A.1, we establish the existence of $L_\psi := L_F + L_{\mathcal{G}} > 0$ such that

$$\|\nabla\psi_{c_k}(x^{k+1}, y^{k+1}, z^{k+1}) - \nabla\psi_{c_k}(x^k, y^k, z^k)\| \le c_k L_\psi \|(x^{k+1}, y^{k+1}, z^{k+1}) - (x^k, y^k, z^k)\|.$$

Using (40) and Lemma A.3, we have

$$\begin{aligned}
\|\nabla\psi_{c_k}(x^k, y^k, z^k) - c_k(d_x^k, d_y^k, d_z^k)\| &= c_k\left\|\nabla\phi_{c_k}(x^k, y^k, z^k) - (d_x^k, d_y^k, d_z^k)\right\| \\
&\le c_k C_\theta\left\|\theta^*(x^k, y^k, z^k) - \theta^{k+1}\right\| \\
&\le c_k C_\theta\left\|\theta^*(x^k, y^k, z^k) - \theta^k\right\|.
\end{aligned} \quad (50)$$

with $C_\theta = (L_f + rL_{g_1} + \frac{1}{\gamma_1} + L_g)^2$. Consequently, we have

$$\|e^k\| \le c_k L_\psi\|(x^{k+1}, y^{k+1}, z^{k+1}) - (x^k, y^k, z^k)\| + \frac{c_k}{\alpha_k}\left\|(x^{k+1}, y^{k+1}, z^{k+1}) - (x^k, y^k, z^k)\right\|$$
$$+ c_k C_\theta\left\|\theta^*(x^k, y^k, z^k) - \theta^k\right\|.$$

Using this bound on $\|e^k\|$, we have that

$$\begin{aligned}
R_k(x^{k+1}, y^{k+1}, z^{k+1}) &\le c_k\left(L_\psi + 1/\alpha_k\right)\|(x^{k+1}, y^{k+1}, z^{k+1}) - (x^k, y^k, z^k)\| \\
&\quad + c_k C_\theta\left\|\theta^*(x^k, y^k, z^k) - \theta^k\right\|.
\end{aligned}$$

Utilizing this inequality, and given that $\alpha_k \in (\underline{\alpha}, c_\alpha)$ for some positive constant $\underline{\alpha}$, we can establish the existence of a constant $C_R > 0$ such that

$$\begin{aligned}
&\frac{1}{c_k^2}R_k(x^{k+1}, y^{k+1}, z^{k+1})^2 \\
&\le C_R\left(\frac{1}{4\alpha_k}\|(x^{k+1}, y^{k+1}, z^{k+1}) - (x^k, y^k, z^k)\|^2 + \frac{\eta C_\theta}{2\gamma_1}\|\theta^k - \theta^*(x^k, y^k, z^k)\|^2\right).
\end{aligned} \quad (51)$$

This inequality, combined with (47), implies that

$$\sum_{k=0}^{\infty}\frac{1}{c_k^2}R_k(x^{k+1}, y^{k+1}, z^{k+1})^2 < \infty. \quad (52)$$

Because $2\rho < 1$, it follows that

$$\sum_{k=0}^{K}\frac{1}{c_k^2} = \sum_{k=0}^{K}\left(\frac{1}{k+1}\right)^{2\rho}\frac{1}{c^2} \ge \left(\int_1^{K+2}\frac{1}{x^{2\rho}}dx\right)\frac{1}{c^2} = \left(\frac{(K+2)^{1-2\rho} - 1}{1-2\rho}\right)\frac{1}{c^2},$$

leading us to conclude from (52) that

$$\min_{0 \le k \le K}R_k(x^{k+1}, y^{k+1}, z^{k+1}) = O\left(\frac{1}{K^{(1-2\rho)/2}}\right).$$

Given that $\psi_{c_k}(x^k, y^k, z^k) \le M$ and $F(x^k, y^k) \ge \underline{F}$ for all $k$, it follows that

$$c_k\mathcal{G}_\gamma(x^k, y^k, z^k) \le M - \underline{F}, \quad \forall k.$$

From $c_k = c(k+1)^\rho$ and $\rho > 0$, we can obtain that

$$\mathcal{G}_\gamma(x^K, y^K, z^K) = O\left(\frac{1}{K^\rho}\right).$$

$\square$

A.6    EXTENSION TO BILEVEL OPTIMIZATION WITH MINIMAX LOWER-LEVEL PROBLEM

In this part, we explore the extension of our proposed gradient-based, single-loop, Hessian-free algorithm, originally designed for bilevel optimization problems with constrained lower-level problems, to bilevel optimization problems with minimax lower-level problem,

$$\min_{x \in X, y \in Y, z \in Z} F(x, y, z) \quad \text{s.t.} \quad (y, z) \in \mathcal{SP}(x),$$

where $\mathcal{SP}(x)$ denotes the set of saddle points for the convex-concave minimax problem,

$$\min_{y \in Y} \max_{z \in Z} f(x, y, z),$$

where $x \in \mathbb{R}^n$, $y \in \mathbb{R}^m$ and $z \in \mathbb{R}^p$, the sets $X$, $Y$ and $Z$ are closed convex sets in $\mathbb{R}^n$, $\mathbb{R}^m$ and $\mathbb{R}^p$, respectively. The UL objective $F : X \times Y \times Z \to \mathbb{R}$, and the LL objective $f : X \times Y \times Z \to \mathbb{R}$ are continuously differentiable with $f$ being convex in $y$ and concave in $z$. Building upon the idea applied in the development of the regularized gap function (3), we introduce the doubly regularized gap function for lower-level minimax problems, defined as:

$$\mathcal{G}_\gamma^{\text{saddle}}(x, y, z) := \max_{\theta \in Y, \lambda \in Z} \left\{ f(x, y, \lambda) - \frac{1}{2\gamma_2} \|\lambda - z\|^2 - f(x, \theta, z) - \frac{1}{2\gamma_1} \|\theta - y\|^2 \right\}.$$

By employing proof techniques analogous to those used in Lemma 2.1 or Theorem 3.3 from Von Heusinger & Kanzow (2009), we can derive similar results for the doubly regularized gap function $\mathcal{G}_\gamma^{\text{saddle}}(x, y, z)$.

**Lemma A.6.** *Assume that $f(x, y, z)$ is convex in $y$ on $Y$ for any given $x \in X, z \in Z$ and concave in $z$ on $Z$ for any given $x \in X, y \in Y$. Let $\gamma_1, \gamma_2 > 0$, we have $\mathcal{G}_\gamma^{\text{saddle}}(x, y, z) \geq 0$ for any $(x, y, z) \in X \times Y \times Z$, and*

$$\mathcal{G}_\gamma^{\text{saddle}}(x, y, z) \leq 0,$$

*if and only if $(y, z) \in \mathcal{SP}(x)$.*

Similarly, utilizing the proof methods in Lemma 2.2, we establish the differentiability of $\mathcal{G}_\gamma^{\text{saddle}}(x, y, z)$ and derive the formula for its gradient.

**Lemma A.7.** *Assume that $f(x, y, z)$ is convex in $y$ on $Y$ for any given $x \in X, z \in Z$ and concave in $z$ on $Z$ for any given $x \in X, y \in Y$ and is continuously differentiable on an open set containing $X \times Y \times Z$. Then $\mathcal{G}_\gamma^{\text{saddle}}(x, y, z)$ is continuously differentiable on $X \times Y \times Z$, and for any $(x, y, z) \in X \times Y \times Z$,*

$$\nabla \mathcal{G}_\gamma^{\text{saddle}}(x, y, z) = \begin{pmatrix} \nabla_x f(x, y, \lambda^*) \\ \nabla_y f(x, y, \lambda^*) \\ -(z - \lambda^*)/\gamma_2 \end{pmatrix} - \begin{pmatrix} \nabla_x f(x, \theta^*, z) \\ (y - \theta^*)/\gamma_1 \\ \nabla_z f(x, \theta^*, z) \end{pmatrix}, \tag{53}$$

*where $\theta^*$ and $\lambda^*$ denote $\theta^*(x, y, z)$ and $\lambda^*(x, y, z)$, respectively, defined as*

$$\theta^*(x, y, z) := \operatorname*{argmin}_{\theta \in Y} \left\{ f(x, \theta, z) + \frac{1}{2\gamma_1} \|\theta - y\|^2 \right\},$$
$$\lambda^*(x, y, z) := \operatorname*{argmax}_{\lambda \in Z} \left\{ f(x, y, \lambda) - \frac{1}{2\gamma_2} \|\lambda - z\|^2 \right\}. \tag{54}$$

This newly introduced gap function enables the following equivalent single-level reformulation of the problem (16),

$$\min_{(x, y, z) \in X \times Y \times Z} F(x, y, z) \quad \text{s.t.} \quad \mathcal{G}_\gamma^{\text{saddle}}(x, y, z) \leq 0.$$

Using Lemma A.6 and the proof techniques in Theorem 2.3, we can establish the equivalence between the reformulation (19) and the bilevel optimization problem (16).

**Theorem A.8.** *Assume that $f(x, y, z)$ is convex in $y$ on $Y$ for any given $x \in X, z \in Z$ and concave in $z$ on $Z$ for any given $x \in X, y \in Y$. Let $\gamma_1, \gamma_2 > 0$, the reformulation (19) is equivalent to the bilevel optimization problem (16).*

Utilizing this reformulation, analogous to BiC-GAFFA, we can propose a gradient-based, single-loop, Hessian-free algorithm for problem (16). At each iteration, we employ a single projected gradient descent step to update $\theta^{k+1}, \lambda^{k+1}$ to approximate $\theta^*(x^k, y^k, z^k)$ and $\lambda^*(x^k, y^k, z^k)$, as follows:

$$\theta^{k+1} = \text{Proj}_Y \left( \theta^k - \eta_k d_\theta^k \right),$$
$$\lambda^{k+1} = \text{Proj}_Z \left( \lambda^k - \eta_k d_\lambda^k \right),$$

where $\eta_k > 0$ is the step size, and

$$
\begin{aligned}
d_\theta^k &:= \nabla_y f(x^k, \theta^k, z^k) + \frac{1}{\gamma_1}(\theta^k - y^k), \\
d_\lambda^k &:= -\nabla_z f(x^k, y^k, \lambda^k) + \frac{1}{\gamma_2}(\lambda^k - z^k).
\end{aligned}
\tag{55}
$$

By substituting $(\theta^{k+1}, \lambda^{k+1})$ for $(\theta^*, \lambda^*)$ in (53), we can approximate the gradients of the function

$$\frac{1}{c} F(x, y, z) + \mathcal{G}_\gamma^{\text{saddle}}(x, y, z)$$

to define the update directions:

$$
\begin{aligned}
d_x^k &:= \frac{1}{c_k} \nabla_x F(x^k, y^k, z^k) + \nabla_x f(x^k, y^k, \lambda^{k+1}) - \nabla_x f(x^k, \theta^{k+1}, z^k), \\
d_y^k &:= \frac{1}{c_k} \nabla_y F(x^k, y^k, z^k) + \nabla_y f(x^k, y^k, \lambda^{k+1}) - (y^k - \theta^{k+1})/\gamma_1, \\
d_z^k &:= \frac{1}{c_k} \nabla_y F(x^k, y^k, z^k) - (z^k - \lambda^{k+1})/\gamma_2 - \nabla_z f(x^k, \theta^{k+1}, z^k).
\end{aligned}
\tag{56}
$$

Finally, we implement an update for the variables $(x, y, z)$ using a step size $\alpha_k > 0$:

$$(x^{k+1}, y^{k+1}, z^{k+1}) = \text{Proj}_{X \times Y \times Z} \left( (x^k, y^k, z^k) - \alpha_k (d_x^k, d_y^k, d_z^k) \right).$$

The complete algorithm is presented in Algorithm 2.

While the primary focus of this paper is the bilevel optimization with constrained lower-level problems, we defer the convergence analysis of this algorithm to future work.

