# OpenReview forum: "Overcoming Lower-Level Constraints in Bilevel Optimization: A Novel Approach with Regularized Gap Functions"
_ICLR.cc/2025/Conference — ICLR 2025 Poster_

### Official Review · Reviewer_CGDJ · 2024-10-19

**Soundness:** 3
**Presentation:** 2
**Contribution:** 2
**Rating:** 6
**Confidence:** 4

**Summary:**

This work studies bilevel problems where the LL problem is constrained.

The key contributions are:

1. The equivalence between main problem and the smoothed problem (6) under convexity.

2. The relation between (6) and the sequence of problems in Thm 3.3

3. A gradient based algorithm with nonasymptotic grad-norm rate for an instance of problem 3.3

Strength:

1.The paper is technically sound.

2.The comparison against known results are clear, and hence the main contributions are clear.

3. I like the discussions on reformulations and the experiments.

Weakness:
I hope the authors could address my concerns below.

1. When and why would constraints in the LL be necessary?

a. GAN would not require constraints.

b. SVM has uncontrained versions.

c. Hyperparameter tuning / data selection can be solved with more naive approach without bilevel problems.

2. There are more than one way to smoothen/regularize the lower level so that the limit is the original problem.

For example: one can add positive coefficients c1, c2, so that the bilevel is now F +  c1 f + c2 max(g, 0) . Then by sending c1, c2/c1 to infinity, one could also get an approximate solution. Why would one formulation better than another?


Minors:
1. The notation N_Y(y) and other similar ones are not defined.

**Strengths:**

See summary

**Weaknesses:**

See summary

**Questions:**

See summary

---

> ### Author Response · Authors · 2024-11-19
>
> > When and why would constraints in the LL be necessary?
> >
> > ​	a. GAN would not require constraints.
> >
> > ​	b. SVM has unconstrained versions.
> >
> > ​	c. Hyperparameter tuning / data selection can be solved with more naive approach without bilevel problems.
>
> **Response**:
>
> Thank you for your questions. We appreciate the opportunity to elaborate on the necessity of constraints in the lower-level (LL) of bilevel optimization problems and address the potential computational challenges.
>
> **When and Why LL Constraints are Necessary**
>
> **Practical Scenarios**: LL constraints are often required to model real-world restrictions and requirements in various applications. For example, in resource allocation, supply chain optimization, and decision analysis, constraints are used to ensure feasible solutions that meet specific conditions such as budget limits, capacity constraints, or physical boundaries.
>
> **Mathematical Model Integrity**: In some problems, LL constraints contribute to a more accurate representation of real-world scenarios. Including constraints helps limit the solution space to realistic and valid outcomes, preventing impractical or suboptimal solutions.
>
> **Stability and Robustness**: LL constraints can provide additional structure to complex optimization problems, including non-convex ones, which enhances solution stability and algorithm robustness. This is particularly beneficial in applications that demand reliable performance under varied conditions.
>
> **Addressing the Specific Examples:**
>
> a. GANs: While it is true that traditional GANs do not explicitly require LL constraints, there are notable examples where incorporating constraints has shown significant benefits. For instance, WGAN-GP [1] and ConGAN [2] introduce constraints in the form of gradient penalties and capacity constraints, respectively, to improve stability and training performance. These constraints are often integrated through penalty functions, which can complicate the optimization process by requiring careful tuning of penalty factors and potentially reducing model interpretability. Our research aims to provide a more systematic approach to handling these constraints effectively.
>
> > [1] Ishaan Gulrajani, Faruk Ahmed, Martin Arjovsky, Vincent Dumoulin, and Aaron C Courville. Improved training of wasserstein gans. In NIPS, 2017
> >
> > [2] Xiaopeng Chao, Jiangzhong Cao, Yuqin Lu, Qingyun Dai, and Shangsong Liang. Constrained generative adversarial networks. Ieee Access, 9:19208–19218, 2021.
>
> b. SVMs: Although there are unconstrained versions of SVMs, constrained SVMs are used in scenarios where additional conditions need to be satisfied for improved robustness and interpretability. For example, constraints may encode prior knowledge about the problem or ensure that the model operates within specific bounds. These constraints can be beneficial in applications where fairness, regulatory compliance, or domain-specific requirements are critical.
>
> c. Hyperparameter Tuning / Data Selection: While more naive approaches may suffice for simple hyperparameter tuning or data selection tasks, bilevel optimization algorithms provide a more efficient and effective framework, especially for complex and high-dimensional problems. Bilevel optimization can handle hierarchical structures where the performance of the upper-level task depends on the optimal solution of the LL problem, resulting in more refined and optimized solutions.

---

> > ### Author Response · Authors · 2024-11-19
> >
> > > There are more than one way to smoothen/regularize the lower level so that the limit is the original problem.
> > > For example: one can add positive coefficients $c_1$, $c_2$, so that the bilevel is now $F + c_1 f + c_2 \max(g, 0)$. Then by sending $c_1$, $c_2/c_1$ to infinity, one could also get an approximate solution. Why would one formulation better than another?
> >
> > **Response**:
> >
> > Thank you for your insightful question. If we understand your concern correctly, you are asking why we do not solve the optimization problem
> > $$
> > \min_{x,y} F(x,y) + c_1 f(x,y) + c_2 \max\\{ g(x,y), 0 \\}
> > $$
> > with sufficiently large values of $c_1$ and $c_2/c_1$ to obtain an approximate solution to the bilevel optimization problem.
> >
> > We would like to clarify that the solution to the optimization problem
> > $$
> > \min_{x,y} F(x,y) + c_1 f(x,y) + c_2 \max\\{ g(x,y), 0 \\}
> > $$
> > does not necessarily converge to the solution of the original bilevel problem as $c_1$ and $c_2/c_1$ goes to infinity. To illustrate this, let us consider the following example of a bilevel optimization problem:
> > $$
> > \begin{aligned}
> > \min_{x \in [-1,1], y \in [-1,1]} \quad & x^2 + y^2 \\
> > \text{s.t.} \quad & y \in \arg\min_{y' \in [-1,1]} xy’.
> > \end{aligned}
> > $$
> > In this case, $X = Y = [-1,1]$, $F(x,y) = x^2 + y^2$, $f(x,y) = xy$, and $g(x,y) = 0$. The solution mapping of the lower-level problem is given by:
> > $$
> > S(x) = \begin{cases}
> > -1, & \text{if } x > 0, \\\\
> > [-1,1], & \text{if } x = 0, \\\\
> > 1, & \text{if } x < 0.
> > \end{cases}
> > $$
> > The optimal solution to this bilevel problem is $(x^*, y^*) = (0,0)$.
> >
> > However, if we consider the approximation problem:
> > $$
> > \min_{x \in [-1,1], y \in [-1,1]} F(x,y) + c_1 f(x,y) = x^2 + y^2 + c_1 xy,
> > $$
> > it can be shown that for sufficiently large $c_1$, the solution to this approximation problem will be $(1,-1)$ and $(-1,1)$. Thus, the minimizers of this approximation problem does not converge to the optimal solution of the bilevel optimization problem $(x^*, y^*) = (0,0)$ as $c_1$ goes to infinity.
> >
> > This example demonstrates that solving the approximation problem
> > $$
> > \min_{x,y} F(x,y) + c_1 f(x,y) + c_2 \max\\{ g(x,y), 0 \\}
> > $$
> > with large values of $c_1$ and $c_2/c_1$ does not necessarily yield an approximation that converges to the solution of the original bilevel optimization problem.
> >
> > > The notation $N_Y(y)$ and other similar ones are not defined.
> >
> > **Response**:
> >
> > Thank you for pointing this out. We will include the definitions of the notation $N_Y(y)$ and other similar terms in the revised version.

---

> > > ### Comment · Reviewer_CGDJ · 2024-11-22
> > > **Thanks**
> > >
> > > I thank the authors for the response and the example.
> > >
> > > I agree that it is hard to preserve the global min of bilevel problems after smoothing/regularization.  Hence, one usually instead study stationary points for some smoothed Lagrangian
> > >
> > > Then I realize that the proposed problem preserves global min under certain conditions (I feel like the bounded z in (7) implies some local sharpness), and hence I think the authors made valid response.
> > >
> > >
> > > In general, I think this is a solid submission, but I still find the defined problem lacks motivation. I think score 6 is currently accurate. I can increase my score if the author gives a more specific problem (or even better, problem class), and explicitly writes out the solutions based on eq(8) showing that the proposed problem definition (R) is better than other alternatives / known definitions.

---

> > > > ### Author Response · Authors · 2024-11-22
> > > >
> > > > We sincerely thank the reviewer for their thoughtful feedback and for providing us with an opportunity to further clarify the motivation behind our proposed reformulation (6) and its associated penalized problem (8).
> > > >
> > > > The primary motivation for introducing this new reformulation comes from the perspective of designing single-loop gradient-based methods for bilevel optimization problems. As Reviewer pgpK also mentioned, there has been a surge of interest in such methods, exemplified by recent works like *"Sow et al., A Primal-Dual Approach to Bilevel Optimization with Multiple Inner Minima"* and *"Ye et al., BOME! Bilevel Optimization Made Easy: A Simple First-Order Approach."* These approaches focus on bilevel problems where the lower-level problem is without coupled constraints. Their algorithmic derivations rely on a value-function-based reformulation of the bilevel optimization problem:
> > > > $$
> > > > \min_{x, y} F(x, y) \quad \text{s.t.} \quad f(x, y) - v(x) \leq 0, \, g(x, y) \leq 0,
> > > > $$
> > > > where $ v(x) := \inf_{y} \\{ f(x, y) \mid g(x, y) \leq 0 \\} $.
> > > > In these works, the smoothness of the value function  $v(x)$ is a critical requirement for establishing the theoretical soundness and convergence of single-loop gradient-based methods.
> > > >
> > > > However, when the lower-level problem involves coupled constraints, the smoothness of $ v(x) $ cannot generally be guaranteed without imposing restrictive assumptions, such as the uniqueness of solutions and Lagrange multipliers for the lower-level problem. For instance, consider the following example:
> > > > $$
> > > > (x, y) \in \mathbb{R} \times \mathbb{R}, \quad f(x, y) = y, \quad g_1(x, y) = x - y \leq 0, \quad g_2(x, y) = -x - y \leq 0.
> > > > $$
> > > > Here, the value function is $ v(x) = |x| $, which is non-differentiable at $ x = 0 $. Consequently, the value-function-based reformulation lacks smoothness, rendering existing single-loop gradient-based methods inapplicable.
> > > >
> > > > This challenge motivates our work. We propose a new smooth doubly regularized gap function for constrained lower-level problems, leading to the smooth single-level reformulation (6). The associated penalized problem (8) inherits this smoothness, enabling us to develop the BiC-GAFFA method, a single-loop gradient-based algorithm. Moreover, we establish its theoretical convergence properties.
> > > >
> > > > In summary, our proposed smooth reformulation provides a new approach for solving bilevel optimization problems with coupled constraints in the lower level. We believe this work can inspire further research in developing effective methods for bilevel optimization problems with constrained lower-level problems.

---

### Official Review · Reviewer_pgpK · 2024-10-25

**Soundness:** 3
**Presentation:** 2
**Contribution:** 3
**Rating:** 6
**Confidence:** 3

**Summary:**

The paper introduces a novel algorithm for constrained bilevel optimization (BiO) using a regularized gap function to transform bilevel problems with lower-level (LL) constraints into a single-level optimization problem with a smooth inequality constraint. The authors propose a first-order, single-loop algorithm, BiC-GAFFA, that operates without requiring Hessian evaluations or projections onto the LL constraint set, offering both theoretical convergence results and empirical validation on synthetic and real-world tasks, such as hyperparameter optimization and generative adversarial networks (GANs). The paper also presents an extension of the method to minimax lower level problems, which broadens its applicability.

**Strengths:**

1. The introduction of the doubly regularized gap function is a novel (as far as I am aware in the context of bilevel optimization) solution to deal with constraints at the lower level in bilevel optimization. This regularization allows the transformation of the bilevel problem into a smooth optimization problem, which simplifies the numerical implementation and analysis.

2. The proposed BiC-GAFFA algorithm avoids costly second-order computations, which is usually the case for methods based on constrained optimization reformulation. The single-loop nature of the algorithm is also computationally efficient, as supported by the non-asymptotic convergence analysis.

3. The extension to minimax inner level problems provides flexibility in dealing with a broader class of problems beyond the traditional settings.

**Weaknesses:**

1. Some theoretical assumptions are insufficiently justified (i.e. very weak statement), particularly in Remark 2.5, where it’s stated that the existence of a multiplier in Theorem 2.3 “can be guaranteed” under certain constraints qualification conditions like MFCQ. Is the assumption in fact guaranteed when the constraints qualification conditions are satisfied? Or does those conditions directly imply that the assumption is satisfied? Please clarify.

2. The update steps in BiC-GAFFA are quite similar to existing constrained optimization reformulation methods, such as primal-dual [1] and dynamic barrier [2] methods. While these approaches have been applied successfully in other constrained settings, the paper does not explain why they are inadequate for the additional lower-level constraint here, suggesting a need for BiC-GAFFA. A discussion on this distinction would help clarify the necessity of more complex methods such as the proposed BiC-GAFFA.

3. The algorithm requires additional sequences (e.g.,  $\theta_k$  and  $\lambda_k$), adding to the computational and memory load, which may challenge its feasibility for large-scale applications.

4. The experimental results focus on toy numerical problems and small-scale real-world problems. While these results are encouraging, a broader evaluation on large-scale or diverse domains would better demonstrate BiC-GAFFA’s practical applicability and scalability.

Reference:

[1] Sow et al. A Primal-Dual Approach to Bilevel Optimization with Multiple Inner Minima.

[2] Ye et al. BOME! Bilevel Optimization Made Easy: A Simple First-Order Approach.

**Questions:**

See weaknesses.

---

> ### Author Response · Authors · 2024-11-19
>
> We would like to thank the reviewer for acknowledging the contributions of our work. Below we would like to give some explanations to clarify the concerns.
>
> > Some theoretical assumptions are insufficiently justified (i.e. very weak statement), particularly in Remark 2.5, where it’s stated that the existence of a multiplier in Theorem 2.3 “can be guaranteed” under certain constraints qualification conditions like MFCQ. Is the assumption in fact guaranteed when the constraints qualification conditions are satisfied? Or does those conditions directly imply that the assumption is satisfied? Please clarify.
>
> **Response**:
>
> Yes, the assumption that "a corresponding multiplier of the lower-level problem (2) exists at $(x,y)$" holds when the constraints qualification conditions, including Guignard’s CQ, Linear Independence Constraint Qualification (LICQ), or Mangasarian-Fromovitz Constraint Qualification (MFCQ), are satisfied for the lower-level problem constraints. This is guaranteed by the established theory for the Karush-Kuhn-Tucker (KKT) conditions. Specifically, when the constraints qualification condition is met, the existence of a KKT point for the constrained optimization problem is ensured, and consequently, the existence of a corresponding multiplier is guaranteed.
>
> > The update steps in BiC-GAFFA are quite similar to existing constrained optimization reformulation methods, such as primal-dual [1] and dynamic barrier [2] methods. While these approaches have been applied successfully in other constrained settings, the paper does not explain why they are inadequate for the additional lower-level constraint here, suggesting a need for BiC-GAFFA. A discussion on this distinction would help clarify the necessity of more complex methods such as the proposed BiC-GAFFA.
>
> **Response**:
>
> Thank you for raising this question. We would like to clarify that both [1] and [2], as well as some other methods in the literature, propose approaches based on the value function reformulation of bilevel optimization problems. Specifically, they consider reformulations of the form:
> $$
> \min_{x, y} F(x, y) \quad \text{s.t.} \quad f(x, y) - v(x) \leq 0, \, g(x, y) \leq 0,
> $$
> where $ v(x) := \inf_{y} \\{f(x, y) \mid g(x, y) \leq 0\\} $. The convergence analysis and theoretical soundness of these single-loop gradient-based methods heavily depend on the smoothness of the value function $ v(x) $ of the lower-level problem.
>
> However, when the lower-level problem includes constraints, the smoothness of $ v(x) $ is generally not guaranteed. For instance, consider the following example:
> $$
> (x, y) \in \mathbb{R} \times \mathbb{R}, \quad f(x, y) = y, \quad g_1(x, y) = x - y \leq 0, \quad g_2(x, y) = -x - y \leq 0.
> $$
>
> In this case, the value function is $ v(x) = |x| $, which is not differentiable at $ x = 0 $. As a result, the value function reformulation of the bilevel optimization problem is not smooth, and the convergence analyses in [1] and [2] do not apply.
>
> To address this challenge, we introduce a new doubly regularized gap function for the constrained lower-level problem. This doubly regularized gap function is smooth, enabling us to propose a smooth, single-level reformulation of the bilevel optimization problem with a constrained lower-level problem. Building on this smooth reformulation, we develop the BiC-GAFFA method, a single-loop gradient-based algorithm, and establish its convergence properties.
>
> Thank you for pointing this out. We will include this discussion in the revised version.

---

> > ### Author Response · Authors · 2024-11-19
> >
> > > The algorithm requires additional sequences (e.g., $\theta_k$ and $\lambda_k$), adding to the computational and memory load, which may challenge its feasibility for large-scale applications.
> >
> > **Response**:
> >
> > Thank you for your valuable feedback. We acknowledge that the introduction of additional sequences, such as $\theta_k$ and $\lambda_k$, does contribute to increased memory usage, which can present challenges for large-scale applications. Addressing this limitation and exploring ways to reduce the storage requirements is an area we plan to investigate in future work.
> >
> > In terms of computational complexity, while our algorithm introduces additional computation, our empirical results suggest that these additional calculations contribute to more effective model optimization. This added complexity, in many cases, proves competitive and can lead to improved convergence and solution quality. Moreover, from the perspective of numerical implementation, our method offers notable advantages over some traditional bilevel optimization algorithms. Unlike conventional approaches, our algorithm does not rely on nested loops or matrix inversion estimations, which significantly enhances computational efficiency. Additionally, when handling lower-level constraints, our method avoids the need for complex projection steps.
> >
> > However, we recognize that large-scale applications of bilevel optimization, particularly those with lower-level constraints, remain in their early stages due to the lack of efficient solution tools. We believe that our work provides a step toward developing more scalable and efficient approaches for these complex problems.
> >
> > > The experimental results focus on toy numerical problems and small-scale real-world problems. While these results are encouraging, a broader evaluation on large-scale or diverse domains would better demonstrate BiC-GAFFA’s practical applicability and scalability.
> >
> > **Response**:
> >
> > Thank you for your insightful feedback. We acknowledge that our current experimental evaluation is primarily focused on toy numerical problems and small-scale real-world scenarios. This limitation was partly due to hardware constraints and the challenge of identifying suitable large-scale models that align with our problem framework.
> >
> >  However, we recognize the importance of demonstrating the broader applicability and scalability of BiC-GAFFA. We are actively working on expanding our research to include more comprehensive evaluations on larger and more diverse datasets. This will allow us to better showcase the practical potential and robustness of our algorithm in real-world, large-scale applications.

---

> > > ### Comment · Reviewer_pgpK · 2024-11-27
> > >
> > > Thank you for the detailed response. Please, the paper should be updated accordingly to reflect the review discussions. At this point, I still believe a score of 6 reflects the quality of this paper, therefore I will keep my score.

---

> > > > ### Author Response · Authors · 2024-11-28
> > > >
> > > > Thank you for your continued feedback. We have updated the manuscript to include the review discussions. Specifically, we have revised the introduction to highlight the motivation to propose a new reformulation, and added a detailed discussion on computational concerns in the appendix. We appreciate your valuable input.

---

### Official Review · Reviewer_ejeH · 2024-11-02

**Soundness:** 3
**Presentation:** 3
**Contribution:** 3
**Rating:** 8
**Confidence:** 5

**Summary:**

The paper introduces a method for transforming constrained bilevel optimization problems with coupled lower-level constraints into a single-level problem with a smooth constraint, employing a doubly regularized gap function. Based on this reformulation, the authors propose a first-order single-loop penalty-based algorithm called BiC-GAFFA. Theoretically, they establish non-asymptotic convergence results under general assumptions. Additionally, they investigate the extension of the proposed algorithm to bilevel optimization problems featuring minimax lower-level problems, a topic that has received limited attention in the existing literature. Extensive experiments are conducted to validate the practicality, robustness and efficiency of BiC-GAFFA, encompassing multiple complex problem models within the learning problems.

**Strengths:**

1.The paper introduces a smooth constrained reformulation for bilevel optimization with a constrained lower-level problem. This reformulation avoids the use of any implicit functions associated with the lower-level problem, which are typically known to necessitate additional iterations or inexact solutions.

2. The authors present the first-order single-loop algorithm BiC-GAFFA, which exhibits overall complexity that is lower than that of many existing first-order algorithms requiring double-loop iterations. Notably, it does not rely on the assumption of full convexity of the lower-level constraints.

3. When updating the proximal variable $\lambda$ associated with the multipliers $z$, there is no need to compute the gradient of the gap function. Instead, the update can be effectively executed solely by applying the first-order optimality conditions of the maximization problem.

4. BiC-GAFFA possesses a broader range of potential applications, making it suitable for solving bilevel optimization problems where the lower-level problem is a minimax problem. Existing bilevel optimization algorithms are unable to achieve this simultaneously.

**Weaknesses:**

1. From Section 6.2 and A.2.2., we observe that when dealing with nonsmooth problems, such as sparse group lasso and SVM, it is necessary to reformulate them into a smooth form for effective resolution. It may impact the applicability of the algorithms.

2. By introducing the gap function, the authors have designed a single-loop algorithm. However, the dimensionality of the introduced variables remains the same as that of the original problem, resulting in an increased iteration scale. Compared to LV-HBA proposed in [1], there is no significant improvement in mathematical representation or convergence results.

3. The necessity to control multiple iteration step sizes within the algorithm significantly affects the convergence theory and empirical performance, making them highly dependent on the adjustment of these step size parameters.

[1] Wei Yao, Chengming Yu, Shangzhi Zeng, and Jin Zhang. Constrained bi-level optimization: Proximal Lagrangian value function approach and Hessian-free algorithm. In ICLR, 2024.

**Questions:**

1. The convergence results for extending the proposed algorithm to bilevel optimization problems with minimax lower-level problems are lacking, which leads to that Section 5 feels somewhat abrupt overall.

2. In the experimental settings, we observe that the step sizes $\alpha_k$ and $\eta_k$ are fixed or vary in a specific form. Are the step sizes related to the specific formulations of different problems?

Some more detailed questions:

1. Can the conclusion of Lemma 2.1 be stated as "$\mathcal{G}_\gamma(x,y,z)=0$ if and only if $y\in S(x)$ and $z\in \mathcal{M}(x, y)$"?

2. In line 845, the loss function in lower-level problem should be $\mathcal{L}_{tr}$.

3. In line 1007, can you provide the specific mathematical format of the functions $\mathcal{L}_{gen/det}$

---

> ### Author Response · Authors · 2024-11-19
>
> We would like to thank the reviewer for acknowledging the contributions of our work. Below we would like to give some explanations to clarify the concerns.
>
> > From Section 6.2 and A.2.2., we observe that when dealing with nonsmooth problems, such as sparse group lasso and SVM, it is necessary to reformulate them into a smooth form for effective resolution. It may impact the applicability of the algorithms.
>
> **Response**:
>
> We acknowledge the challenges posed by nonsmooth functions in optimization problems. Addressing these challenges is a valuable direction for future research, and we plan to explore strategies for effectively handling nonsmooth functions in subsequent studies.
>
> While our theoretical convergence guarantees rely on the assumption of smoothness for the functions involved, this does not preclude the application of our method in scenarios where smoothness conditions are not fully satisfied. In practice, for a wide range of real-world applications, automatic differentiation often provides an effective approximation or substitution for subgradients, enabling the computation of update directions to proceed effectively. Developing a rigorous theoretical framework to support these practical extensions is an important area for future work.
>
> > By introducing the gap function, the authors have designed a single-loop algorithm. However, the dimensionality of the introduced variables remains the same as that of the original problem, resulting in an increased iteration scale. Compared to LV-HBA proposed in [1], there is no significant improvement in mathematical representation or convergence results.
>
> **Response**:
>
> We would like to clarify that, compared to the LV-HBA method in [1], the proposed algorithm in this work offers significant advantages. Specifically, it eliminates two key restrictions present in LV-HBA. First, it relaxes the requirement for joint convexity of the lower-level constraint $ g(x, y) \leq 0 $ with respect to $(x,y)$. Second, it removes the need to compute the projection onto the set $\\{(x, y) \mid g(x, y) \leq 0 \\} $. Instead, the proposed method requires only the convexity of the lower-level constraint $ g(x, y) \leq 0 $ with respect to the lower-level variable $ y $, and does not require the costly projection operation onto the set $ \\{(x, y) \mid g(x, y) \leq 0 \\} $. These two improvements significantly broaden the applicability of the proposed method to real-world problems, compared to LV-HBA.
>
> > The necessity to control multiple iteration step sizes within the algorithm significantly affects the convergence theory and empirical performance, making them highly dependent on the adjustment of these step size parameters.
>
> **Response**:
>
> We would like to clarify that our proposed algorithm involves only two step sizes, $ \eta_k $ and $ \alpha_k $. Regarding the convergence theory, the requirements for these step sizes are relatively mild. Specifically, the convergence results hold as long as the step sizes $ \eta_k $ and $ \alpha_k $ are chosen to be small constants.
>
> In practical implementations, while step sizes indeed play a critical role in the empirical performance of the algorithm, we have found that initial values of $\alpha_k = 0.0001$ and $\eta_k = 0.001$ typically provide a good starting point. These values can be adjusted based on the specific problem setting to optimize performance. For example, in scenarios with faster or slower convergence behavior, the step sizes can be fine-tuned accordingly to balance convergence speed and algorithmic stability.
>
> Moreover, adaptive step size strategies, such as using techniques inspired by learning rate schedulers or gradient normalization methods, could further enhance practical performance and reduce sensitivity to initial choices.
>
>
>
> > The convergence results for extending the proposed algorithm to bilevel optimization problems with minimax lower-level problems are lacking, which leads to that Section 5 feels somewhat abrupt overall.
>
> **Response**:
>
> We would like to clarify that Section 5 is included to highlight the potential of the doubly regularized gap function idea, demonstrating its applicability not only to bilevel optimization problems with constrained lower-level problems, but also to more complex problems such as bilevel optimization with minimax lower-level optimization. The aim was to show that the proposed technique has broader applicability and can lead to a smooth single-level reformulation for minimax bilevel optimization. However, since the primary focus of this paper is on bilevel optimization with constrained lower-level problems, we chose to provide only a brief discussion on the minimax case. A more detailed analysis of bilevel optimization problems with minimax lower-level problems will be explored in future work.

---

> ### Author Response · Authors · 2024-11-19
>
> > In the experimental settings, we observe that the step sizes  $\alpha_k$ and $\eta_k$ are fixed or vary in a specific form. Are the step sizes related to the specific formulations of different problems?
>
> **Response**:
>
> Thank you for your question. We appreciate the opportunity to clarify the step size selection in our experimental settings. On one hand, our theoretical convergence analysis indicates that sufficiently small step sizes are adequate to ensure the convergence of our algorithm. To simplify the practical application of our method and reduce the tuning burden, we chose $\alpha_k = \eta_k/10$ as the default step size relationship. This choice was based on its strong empirical performance across various tests, and we recommend it as an initial step size configuration for users. However, it is important to note that this step size relationship is not a strict requirement. Users can adjust $\alpha_k$ and $\eta_k$ according to the specific characteristics and needs of different problems.
>
> > Can the conclusion of Lemma 2.1 be stated as "$\mathcal{G}_\gamma (x,y,z)=0$ if and only if $y\in S(x)$ and $z\in M(x,y)$"?
>
> **Response**:
>
> Yes, the conclusion of Lemma 2.1 can be stated as $ \mathcal{G}_ \gamma(x,y,z) = 0 $ if and only if $ y \in S(x) $ and $ z \in \mathcal{M}(x, y) $. This is because $ \mathcal{G}_ \gamma(x, y, z) \geq 0 $ for any $ (x, y, z) \in X \times Y \times \mathbb{R}_ +^p $ (as shown in Lemma 2.1), and thus the constraint $ \mathcal{G}_ \gamma(x, y, z) \leq 0 $ defines the same feasible set as the constraint $ \mathcal{G}_ \gamma(x, y, z) = 0 $ on $ (x, y, z) \in X \times Y \times \mathbb{R}_ +^p $.
>
> > In line 845, the loss function in lower-level problem should be $\mathcal{L}_{tr}$.
>
> Thank you for pointing out this oversight. We have corrected the expression to the following:
> $$
> \min_{\mathbf{x} \in \mathbb{R}^n, \mathbf{r} \in \mathbb{R}^J} \mathcal{L}_ {\text{val}}(\mathbf{x})
> \\ \text{ s.t. }\\ \mathbf{x} \in  \mathop{\mathrm{argmin}}_ {\mathbf{x} \in \mathbb{R}^n} \Big\\{
> \mathcal{L}_ {\text{tr}}(\mathbf{x})  \\ \text{ s.t. }\ P_i(\mathbf{x}) \le r_ i,\ i=1,\cdots,J
> \Big\\}.
> $$
> We appreciate your attention to this detail.

---

> ### Author Response · Authors · 2024-11-19
>
> > In line 1007, can you provide the specific mathematical format of the functions $\mathcal{L}_{gen/det}$.
>
> Thank you for your feedback. We appreciate your attention to detail. In our experiments, we used the following specific mathematical formats for the loss functions $\mathcal{L}{gen}$ and $\mathcal{L}{det}$:
>
> 1. GAN:
> $$
> \begin{aligned} \mathcal{L}_ {\text{gen}}(G, D) = & \mathbb{E}_ {\mathbf{z}\sim \mathbb{N}}[\mathrm{BCELoss}(D(G(\mathbf{z})), \mathbb{1}_ n)],\\\\ \mathcal{L}_ {\text{det}}(G, D) = & \frac12\mathbb{E}_ {\mathbf{x}\sim \mathbb{P}_ r}[\mathrm{BCELoss}(D(\mathbf{x}), \mathbb{1}_  n)] +\frac12 \mathbb{E}_ {\mathbf{z}\sim \mathbb{N}}[\mathrm{BCELoss}(D(G(\mathbf{z})), \mathbb{0}_ n)]. \end{aligned}
> $$
>
> 2. WGAN:
> $$
> \begin{aligned} \mathcal{L}_ {\text{gen}}(G, D) = & -\mathbb{E}_ {\mathbf{z}\sim \mathbb{N}}[D(G(\mathbf{z})],\\\\
> \mathcal{L}_ {\text{det}}(G, D) = & - \mathbb{E}_ {\mathbf{x}\sim \mathbb{P}_ r}[D(\mathbf{x})] +\mathbb{E}_ {\mathbf{z}\sim \mathbb{N}}[D(G(\mathbf{z})], \end{aligned}
> $$
> with a box constraint that $\\|D\\|_ {\infty} \le c_ \mathrm{clip}$.
>
> 3. WGAN-GP:
> $$
> \begin{aligned}
> \mathcal{L}_ {\text{gen}}(G, D) = & -\mathbb{E}_ {\mathbf{z}\sim \mathbb{N}}[D(G(\mathbf{z})],\\\\ \mathcal{L}_ {\text{det}}(G, D) = & - \mathbb{E}_ {\mathbf{x}\sim \mathbb{P}_ r}[D(\mathbf{x})]+\mathbb{E}_ {\mathbf{z}\sim \mathbb{N}}[D(G(\mathbf{z})] + \lambda_ {\text{gp}} \mathbb{E}_ {\hat{\mathbf{x}}\sim \mathbb{P}_ {\hat{\mathbf{x}}}}(\|\nabla D(\hat{\mathbf{x}})\|_ 2 - 1)^2,
> \end{aligned}
> $$
> where $\lambda_ {\text{gp}}$ is the parameter of gradient penalty, and $\mathbb{P}_ {\hat{\mathbf{x}}}$ is defined by sampling uniformly along straight lines between pairs of points sampled from the data distribution $\mathbb{P}_ r$ and the generator distribution $\mathbb{P}_ g := \\{G(\mathbb{z}) | \mathbb{z}\in \mathbb{N}\\}$.
>
> 4. Con-GAN:
> $$
> \begin{aligned}
> \mathcal{L}_ {\text{gen}}(G, D) = & \mathbb{E}_ {\mathbf{z}\sim \mathbb{N}}[\mathrm{BCELoss}(D(G(\mathbf{z})), \mathbb{1}_ n)],\\\\ \mathcal{L}_ {\text{det}}(G, D) = & \mathbb{E}_ {\mathbf{x}\sim \mathbb{P}_ r}[\mathrm{BCELoss}(D(\mathbf{x}), \mathbb{1}_ n)] +\mathbb{E}_ {\mathbf{z}\sim \mathbb{N}}[\mathrm{BCELoss}(D(G(\mathbf{z})), \mathbb{0}_ n)] + \lambda h(G, D)
> \end{aligned}
> $$
> where
> $$
> h(G, D) := \mathbb{E}_{\{\mathbf{x}\sim \mathbb{P}_r, \mathbf{z}\sim \mathbb{N}\}} [\log(D(\mathbf{x})) - \log D(G(\mathbf{z}))]^2.
> $$
>
> 5. UGAN:
> $$
> \begin{aligned}
> \mathcal{L}_ {\text{gen}}(G, D) = & \mathbb{E}_ {\mathbf{z}\sim \mathbb{N}}[\mathrm{BCELoss}(D(G(\mathbf{z})), \mathbb{1}_ n)],\\\\
> \mathcal{L}_ {\text{det}}(G, D) = & \frac12\mathbb{E}_ {\mathbf{x}\sim \mathbb{P}_ r}[\mathrm{BCELoss}(D(\mathbf{x}), \mathbb{1}_ n)] +\frac12 \mathbb{E}_ {\mathbf{z}\sim \mathbb{N}}[\mathrm{BCELoss}(D(G(\mathbf{z})), \mathbb{0}_ n)].
> \end{aligned}
> $$
>
> 6. Bi-GAN:
> $$
> \begin{aligned}
> \mathcal{L}_ {\text{gen}}(G, D) = & \mathbb{E}_ {\mathbf{z}\sim \mathbb{N}}[\mathrm{BCELoss}(D(G(\mathbf{z})), \mathbb{1}_ n)],\\\\ \mathcal{L}_ {\text{det}}(G, D) = & \frac12\mathbb{E}_ {\mathbf{x}\sim \mathbb{P}_ r}[\mathrm{BCELoss}(D(\mathbf{x}), \mathbb{1}_  n)]+\frac12 \mathbb{E}_ {\mathbf{z}\sim \mathbb{N}}[\mathrm{BCELoss}(D(G(\mathbf{z})), \mathbb{0}_ n)]. \end{aligned}
> $$
>
> 7. Bi-WGAN:
> $$
> \begin{aligned} \mathcal{L}_ {\text{gen}}(G, D) = & -\mathbb{E}_ {\mathbf{z}\sim \mathbb{N}}[D(G(\mathbf{z})],\\\\
> \mathcal{L}_ {\text{det}}(G, D) = & - \mathbb{E}_ {\mathbf{x}\sim \mathbb{P}_ r}[D(\mathbf{x})] +\mathbb{E}_ {\mathbf{z}\sim \mathbb{N}}[D(G(\mathbf{z})], \end{aligned}
> $$
> with constraint
> $$
> \max_ {\hat{\mathbf{x}}\in \mathbb{P}_ {\hat{\mathbf{x}}}} \|\nabla D(\hat{\mathbf{x}})\|_ 2 \le 1
> $$
> where  $\mathbb{P}_ {\hat{\mathbf{x}}}$ is defined by sampling uniformly along straight lines between pairs of points sampled from the data distribution $\mathbb{P}_ r$ and the generator distribution $\mathbb{P}_ g := \\{G(\mathbb{z}) | \mathbb{z}\in \mathbb{N}\\}$.
>
> 8. Bi-ConGAN:
> $$
> \begin{aligned}
> \mathcal{L}_ {\text{gen}}(G, D) = & \mathbb{E}_ {\mathbf{z}\sim \mathbb{N}}[\mathrm{BCELoss}(D(G(\mathbf{z})), \mathbb{1}_ n)],\\\\
> \mathcal{L}_ {\text{det}}(G, D) = & \mathbb{E}_ {\mathbf{x}\sim \mathbb{P}_ r}[\mathrm{BCELoss}(D(\mathbf{x}), \mathbb{1}_ n)] +\mathbb{E}_ {\mathbf{z}\sim \mathbb{N}}[\mathrm{BCELoss}(D(G(\mathbf{z})), \mathbb{0}_ n)]
> \end{aligned}
> $$
> with constraint
> $$
> h(G, D) := \mathbb{E}_ {\\{\mathbf{x}\sim \mathbb{P}_r, \mathbf{z}\sim \mathbb{N}\\}} [\log(D(\mathbf{x})) - \log D(G(\mathbf{z}))]^2 \le \epsilon.
> $$
>
> We will incorporate these detailed expressions in the revised manuscript for clarity and completeness.

---

> ### Comment · Reviewer_ejeH · 2024-11-24
>
> Thanks for your patient response. We have increased the rating to 8. We also looks forward to your more detailed analysis of bilevel optimization problems with minimax lower-level problems to be explored in future work.

---

> > ### Author Response · Authors · 2024-11-25
> >
> > Thank you for your thoughtful feedback and for increasing the rating. We truly appreciate your recognition of our work and your encouragement regarding our exploration of minimax problems.

---

### Official Review · Reviewer_qRB2 · 2024-11-04

**Soundness:** 3
**Presentation:** 3
**Contribution:** 3
**Rating:** 6
**Confidence:** 4

**Summary:**

This paper studies a bilevel problem where lower-level problem is convex with coupled constraints. To avoid joint projection onto the coupled constraint set which requires coupling convexity and is costly, this paper introduces a doubly regularized gap function to convert lower-level problem into a smooth constraint. This constrained problem is then solved by penalty method. Convergence analysis is provided and numerical results validate the effectiveness of the proposed method.

**Strengths:**

1. This work tackles an important bilevel problem which involves coupled constraints at the lower level.
2. The design of the doubly regularized gap function is novel and effective.
3. The numerical studies are comprehensive and thorough.

**Weaknesses:**

This paper addresses a relaxation of the original bilevel problem through a penalty reformulation and truncation approach. For the penalty reformulation, it is shown that by selecting a sufficiently large penalty constant $c$, the penalty problem becomes an $\epsilon$-approximate solution to the original problem. At the same time, the truncation parameter $r$ also appears to need a large value to ensure equivalence with the original problem. But in the final convergence theorem, $r$ is chosen as a constant, independent of the target error $\epsilon$.

**Questions:**

In Proposition 3.1, it is assumed that the optimal solution $(x^*,y^*,z^*)$ to (6) exists with finite $z^*$. How can this be guaranteed? Additionally, how should $r$ be chosen in practice? Setting $r$ too large may hinder algorithm convergence, while choosing $r$ too small could exclude the optimal point $z^*$, which is unknown.

Additionally, several concurrent related works are missing: [1]--[2].

[1] A Primal-Dual-Assisted Penalty Approach to Bilevel Optimization with Coupled Constraints. L Jiang, et. al. arXiv:2406.10148.

[2] First-Order Methods for Linearly Constrained Bilevel Optimization. G Kornowski, et.al. 	arXiv:2406.12771.

In Table 3, the term "Required accuracy" could be replaced with a word like "Time" to clarify the values reported. As it stands, it is unclear whether the numbers in this table represent accuracy or computation time.

---

> ### Author Response · Authors · 2024-11-19
>
> > “This paper addresses a relaxation ...... independent of the target error $\epsilon$."
>
> **Response**: We would like to clarify that the choice of $r$ as a constant in the convergence theorem is based on the reasoning presented in Proposition 3.1. Specifically, once $r$ is appropriately selected to be sufficiently large—such that it encompasses at least one Lagrange multiplier $z^*$ of the lower-level problem at the optimal solution point $(x^*, y^*)$, satisfying $\|z^*\|_{\infty} \leq r$ —the optimal solution of the truncated variant (7) also becomes optimal for the original reformulation (6). Therefore, in the convergence theorem, we focus on the truncated variant (7) with this appropriately chosen $ r $.
>
> > “In Proposition 3.1, it is assumed that the optimal solution $(x^*, y^*, z^*)$ to (6) exists with finite $z^*$. How can this be guaranteed? Additionally, how should r be chosen in practice? Setting $r$ too large may hinder algorithm convergence, while choosing $r$ too small could exclude the optimal point $z^*$, which is unknown."
>
> **Response:**
>
> To clarify, for any optimal solution $ (x^*, y^*, z^*)$ of (6), $z^*$ corresponds to a Lagrange multiplier associated with the lower-level problem at the optimal solution point $ (x^*, y^*) $. By definition, such Lagrange multiplier is finite if it exists. Furthermore, the existence of multiplier is guaranteed under standard constraint qualification conditions for the lower-level problem constraints, as discussed in Remark 2.5. We will include additional explanations around Proposition 3.1 in the revised manuscript to make this clearer.
>
> Regarding the selection of the truncation parameter $ r $, theoretically, $ r $ should be chosen large to ensure it includes at least one feasible Lagrange multiplier $ z^* $. In practice, $ r $ can be tuned based on problem-specific characteristics or adjusted adaptively during the algorithm’s execution. Specifically, the parameter $ r $ can be selected as follows: Initially, a relatively large $ r $ can be chosen, and the behavior of $ F $, $ f $, and $\max(z)$ can be monitored during computations. If the results exhibit stable convergence characteristics, the chosen $ r $ can be retained. Conversely, if significant oscillations or instability are observed, $ r $ can be adjusted iteratively based on prior observations of $\max(z)$. Furthermore, grid search provides a systematic approach for exploring different $ r $ values to identify the optimal balance between convergence stability and computational performance.
>
> > Additionally, several concurrent related works are missing: [1]--[2].
> >
> > [1] A Primal-Dual-Assisted Penalty Approach to Bilevel Optimization with Coupled Constraints. L Jiang, et. al. arXiv:2406.10148.
> >
> > [2] First-Order Methods for Linearly Constrained Bilevel Optimization. G Kornowski, et.al. arXiv:2406.12771.
>
> **Response**: We thank the reviewer for pointing out these recent and relevant references. We will include a discussion of these works in the revised manuscript.
>
> > In Table 3, the term "Required accuracy" could be replaced with a word like "Time" to clarify the values reported. As it stands, it is unclear whether the numbers in this table represent accuracy or computation time.
>
> **Response**:
>
> Thank you for your observation regarding Table 3. We appreciate the opportunity to clarify this. The numbers in the table represent the computation time required to achieve the target accuracy, as indicated by the metric EM < 0.5 and EM < 0.1. To avoid any confusion, we agree that changing the column heading from "Required accuracy" to "Time to reach target accuracy" would better reflect the content of the table. We will make this change in the revised manuscript to improve clarity.

---

> > ### Comment · Reviewer_qRB2 · 2024-11-27
> >
> > Thank the author for the detailed response. It solves all of my concerns so that I will keep a positive score.

---

### Meta-Review · Area_Chair_DgDZ · 2024-12-20

**Metareview:**

This paper studies bilevel optimization problems with coupled lower-level constraints beyond linear lower-level constraints. This is a challenging setting because the standard bilevel methods do not work. The novelty of this paper is that it introduces a new penalty function and the corresponding gradient update to solve the resultant problems.  This paper addresses an important problem and provide a comprehensive answer. Therefore, I will support for "accept."

**Additional Comments On Reviewer Discussion:**

During the rebuttal period, all the reviewers actively engaged with the author, and some minor questions were clarified.

---

### Decision · Program_Chairs · 2025-01-22

Accept (Poster)